# Sulfatase modifying factors control the timing of zebrafish convergence and extension morphogenesis

Ailen Soledad Cervino[1,2], Amrita Basu[3], Ryan J. Weiss ●[3,4], Gursimran Kaur Bajwa[5,6], Rubén Marín-Juez[5,6], Sandra L. Grimm ●[1,2], Cristian Coarfa[1,2] & Margot Kossmann Williams ●[1,2] ✉

Convergence and extension (C&E) cell movements that elongate the primary embryonic axis are precisely timed during vertebrate gastrulation, but mechanisms controlling their onset remain unknown. Using zebrafish embryonic explants that recapitulate C&E and its timing, we identified *sulfatase modifying factor 2* (*sumf2*) as a candidate trigger gene for C&E onset. *sumf2* and its paralog *sumf1* encode negative and positive sulfatase regulators, respectively, whose expression levels invert and increase heparan sulfate sulfation during gastrulation. Overexpressing *sumf1* or *sumf2* causes delayed or precocious C&E, respectively, whereas their loss shifts C&E timing in the opposite direction. We identified Sulf1, a modifier of heparan sulfate proteoglycans (HSPGs), as their key downstream effector and found that altering heparan sulfate sulfation levels shifts C&E onset and suppresses *sumf1* and *sumf2* mutant phenotypes. This work supports a model in which *sumf2* expression reduces sulfatase activity, rewriting HSPG sulfation patterns to promote the onset of C&E morphogenesis.

Morphogenetic cell movements must be coordinated not only in space, but also in time to properly shape the embryonic body plan. Indeed, changes in the timing of developmental events, termed heterochrony, can underlie malformations in individuals and fuel evolutionary change in populations[1,2]. One striking example of morphogenetic timing is the onset of gastrulation cell movements that form the primordial germ layers and shape them into the nascent embryonic axes. In many species, multiple gastrulation movements occur simultaneously and/or in rapid succession, such that the absolute timing of one process is necessary to preserve its timing relative to the others. For example, many teleost fish and amphibian embryos exhibit epiboly (which thins and spreads the epiblast), internalization (which brings mesoderm and endoderm germ layers inside the embryo), and convergence & extension (C&E) (which elongate the anteroposterior axis) movements simultaneously. The onset of epiboly in zebrafish is thought to result from a rapid fluidization of the epiblast, a consequence of cell rounding during meta-synchronous cell cleavages[3,4]. Initial internalization of mesoderm and endoderm cells may be triggered by a threshold level of Nodal signaling activity[5], which promotes cell protrusions that un-jam cells to enable them to ingress at the margin[6] or ectopically[7]. Once leader cells ingress at the margin, followers in adjacent spatial domains subsequently internalize in a temporally ordered fashion according to their expression of *hox* genes[8], similar to a mechanism reported during chick gastrulation[9]. However, mechanisms controlling the timing of C&E morphogenesis remain elusive.

C&E movements simultaneously narrow and elongate embryonic tissues via polarized cell rearrangements, providing the major driving

[1]Center for Precision Environmental Health, Baylor College of Medicine, Houston, TX, USA. [2]Department of Molecular & Cellular Biology, Baylor College of Medicine, Houston, TX, USA. [3]Complex Carbohydrate Research Center, University of Georgia, Athens, GA, USA. [4]Department of Biochemistry and Molecular Biology, University of Georgia, Athens, GA, USA. [5]Centre de Recherche Azrieli, Centre Hospitalier Universitaire Sainte-Justine, Montréal, QC, Canada. [6]Department of Pathology and Cell Biology, Faculté de Médecine, Université de Montréal, Montréal, QC, Canada. ✉e-mail: margot.williams@bcm.edu

force of anteroposterior (AP) axis extension and, in many species, neural tube closure[10–13]. C&E are driven by polarized cell behaviors including mediolateral (ML) cell elongation and alignment, ML-biased cell protrusions, and polarized contraction of cell interfaces by which cells exchange neighbors to form a longer and narrower array[14,15]. Vertebrate C&E movements and/or their underlying ML intercalation behaviors often begin at mid-gastrulation[16–18]. In zebrafish, this is marked by a switch in the trajectory of cell movements toward the embryo's dorsal side at ~75% epiboly (~8 hours post fertilization (hpf))[18,19]. Evidence suggests that these movements do not result from chemotaxis[20], but rather from spatially restricted cell-cell interactions whose domains are defined by morphogen signaling gradients[21–23].

The signature ML polarity of vertebrate C&E behaviors is under the control of planar cell polarity (PCP) signaling, which orients cell behaviors with respect to the embryonic axes[24,25]. This results from the polarized membrane localization of core PCP signaling components[26–29], which directly precedes the start of C&E[22,30]. The precision and coordination of these events imply the existence of a timing cue that determines the onset of C&E cell behaviors. Several morphogen signaling pathways are essential for C&E, with BMP preventing C&E of the ventral mesoderm[21,31,32] and Nodal and FGF signaling promoting C&E dorsally[22,33–37]. However, all these pathways are active hours before C&E begins at mid-gastrulation, suggesting that their activity might not be directly responsible for the timing of C&E onset. Indeed, although Activin (which signals through the Nodal pathway) was sufficient to induce axial mesoderm exhibiting C&E in *Xenopus* animal cap explants, the timing of C&E remained constant regardless of when Activin was applied[38]. This suggests that the timing mechanism functions in parallel with known C&E regulators, but its molecular basis is unknown.

Mechanisms controlling the timing of other early embryonic events have been extensively studied, including early cell cycles and zygotic genome activation (ZGA). For example, early cell cleavages are controlled by cyclical expression and degradation of maternally expressed Cyclins[39]. After a set number of cell divisions, ZGA is triggered by a threshold nuclear/cytoplasmic (N/C) ratio[40,41–43] which is thought to function through titration of histones and other nuclear factors[42–45]. In amphibian embryos, the timing of gastrulation was linked not to fertilization or ZGA, but to the first embryonic cleavage[46,47], and this timing is reportedly regulated predominantly by cytoplasmic factors[46,48–50]. However, the onset of gastrulation morphogenesis is unchanged in embryos with varying cell size and cell cycle length[50–53], making mechanisms that count cell cycles or measure N/C ratios unlikely regulators of C&E timing. Zebrafish gastrulation morphogenesis requires zygotic transcription[54], but its timing is uncoupled from that of ZGA, instead supporting a model by which new gene expression triggers C&E at a specific time.

Using a reductive zebrafish embryonic explant model in which C&E is isolated from the other gastrulation cell movements, we determined that new transcription is required at gastrulation onset for C&E to occur, and that *sulfatase modifying factor 2* (*sumf2*) is initially expressed during this critical window in both explants and intact embryos. *sumf1* and *sumf2* encode Formylglycine Generating Enzyme (FGE)[55–58] and its antagonist and paralog pFGE[59–61], respectively, a pair whose balance determines the activity of every sulfatase enzyme in the body[62–65]. *sumf1* is maternally expressed, and its transcript abundance drops just as *sumf2* is expressed at gastrulation onset, inverting *sumf1/ sumf2* levels and altering sulfation in the embryo. We show that overexpression of *sumf1* and *sumf2* causes delayed or precocious C&E onset, respectively, in both explants and embryos, whereas loss of *sumf1* and *sumf2* function shifts C&E timing in the opposite direction. We further identified Sulf1 as a key sulfatase by which FGE (*sumf1*) and pFGE (*sumf2*) modify C&E timing. Reduced or increased levels of sulfated heparan sulfate, the predominant substrate of Sulf1[66–68], similarly shift C&E onset and suppress *sumf1* and *sumf2* mutant phenotypes,

indicating that heparan sulfate proteoglycans (HSPGs) mediate C&E timing downstream of sulfatase modifiers. Together, these data support a model in which *sumf2* expression at gastrulation onset reduces sulfatase activity, triggering a switch in HSPG sulfation patterns to promote and/or permit C&E morphogenesis.

## Results

### Ex vivo convergence & extension requires new transcription at gastrulation onset

Blocking transcription prior to ZGA prevents all gastrulation morphogenesis in zebrafish embryos[54]. However, it was not determined when new transcription is required for gastrulation cell movements nor the specific genes that are required. We previously showed that otherwise naïve zebrafish embryonic explants expressing Nodal signaling components recapitulate both C&E behaviors and their precise timing in culture, while lacking other gastrulation movements[36], making them an ideal system to study C&E timing. To identify the temporal window of gene expression required for C&E, we treated explants expressing the constitutively active Nodal receptor Acvr1b* (also known as TARAM-A-D[69]) with a time-course of the irreversible transcription inhibitor Triptolide[70]. We then assessed their ability to undergo C&E by measuring their extension when sibling embryos reached the 4-somite stage (12 hpf) (Fig. 1A). *acvr1b** explants treated when intact siblings reached 50% epiboly (5.3 hpf) failed entirely to extend, while those treated shortly thereafter at shield stage (6 hpf, gastrulation onset) or later were able to extend, albeit incompletely (Fig. 1B, C).

To rule out the possibility that Triptolide interferes with C&E by disrupting Nodal signaling dynamics, we repeated the experiment by activating Nodal under two additional experimental conditions. First, we expressed the Nodal ligand Ndr2/Cyc in explants, which display delayed Nodal activation compared to *acvr1b** explants[71]. Second, we expressed *acvr1b** in explants generated from Nodal signaling-deficient maternal-zygotic (MZ)*oep*-/- embryos[72], which exhibit continued signaling in the absence of feed-forward synthesis of new Nodal ligands. Both conditions responded identically to wildtype (WT) *acvr1b** explants, indicating that the requirement for transcription is independent of Nodal signaling per se (Supplementary Fig. 1A, B, D, E). This is consistent with Activin-induced *Xenopus* animal cap explants extending on time regardless of when in the competence window Activin was added[38]. Finally, WT *acvr1b** explants treated with the reversible transcription inhibitor Flavopiridol[73] at 50% epiboly and washed out at 90% epiboly (5.3–9 hpf) similarly failed to extend (Supplementary Fig. 1C, F), demonstrating that the stage of inhibition, rather than its duration, was responsible for failed C&E. These results indicate that new gene expression starting at gastrulation onset is required for ex vivo C&E movements, independent of Nodal signaling dynamics.

### *sumf1* and *sumf2* transcript levels invert at gastrulation onset

Having established that transcription at gastrulation onset is crucial for C&E, we next identified genes whose expression is upregulated during this time window. We previously profiled transcription in *ndr2, acvr1b**, and uninjected control explants by bulk RNA sequencing at seven developmental stages spanning gastrulation[71] (Fig. 1D). We examined these data for candidate trigger genes whose expression remained unchanged between sphere and 50% epiboly stages but then increased significantly at shield stage and beyond, corresponding to our experimentally determined transcriptional window. Because the timing of C&E onset is independent of Nodal signaling dynamics, we only considered genes with this temporal expression profile in explants of all three conditions, including uninjected controls. This analysis yielded a list of 180 genes, 129 of which also increased significantly between 50% epiboly and shield stages in intact zebrafish gastrulae[74] (Supplementary Data 1).

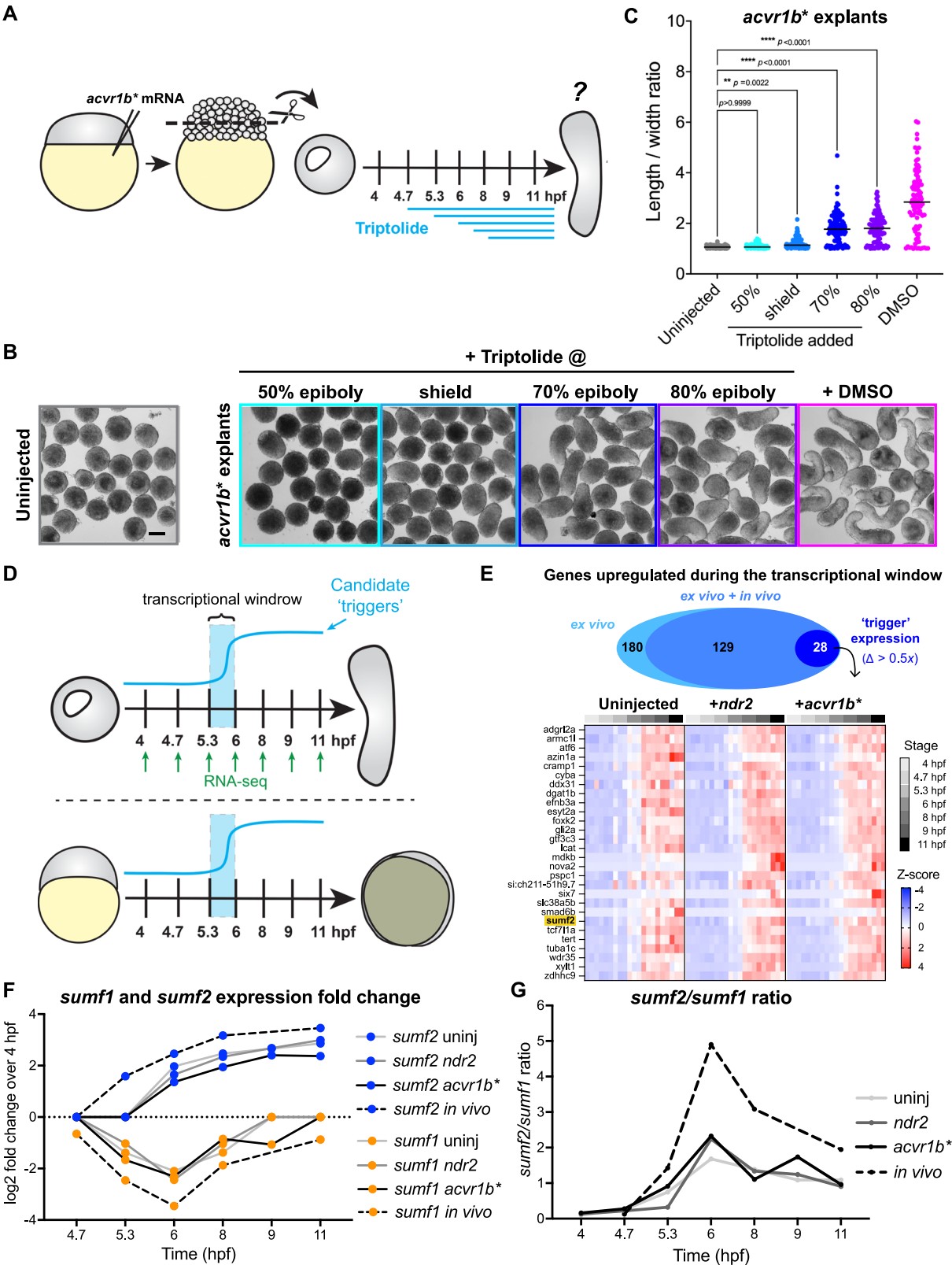

**C** *acvr1b\* explants*

**B** + Triptolide @

**D** transcriptional windrow — Candidate 'triggers'

**E** Genes upregulated during the transcriptional window

**F** *sumf1 and sumf2 expression fold change*

**G** *sumf2/sumf1 ratio*

To focus our efforts on genes with the highest likelihood of functional relevance, we further selected for those exhibiting substantial increases in expression and overall expression levels. To this end, we selected an arbitrary cut-off of a 50% increase ($\Delta > 0.5x$) in transcript level from 50% epiboly to shield stage, and an expression level of at least 5 transcripts per million (TPM) at shield stage in intact embryos. When narrowed down this way, we were left with 28 candidate genes (Fig. 1E)

(Supplementary Data 1) that we hypothesized may trigger C&E cell behaviors when expressed at gastrulation onset. They encode proteins with a wide variety of biological functions, including transcription and chromatin factors (*atf6, cramp1, gtf3c3, six7, tcf7l1a*), cytoskeletal regulation (*tuba1c, wdr35*), cell signaling (*adgrl2a, efnb3a, gli2a, mdkb, smad6b*), biosynthetic processes (*azin1a, dgat1b, lcat, xylt1*), RNA binding (*ddx31, nova2, pspc1*), and others (Supplementary Data 1).

**Fig. 1 | Ex vivo convergence & extension requires new transcription at gastrulation onset. A** Diagram of zebrafish embryonic explants and Triptolide treatments (modified from[36]). All explants were generated from WT embryos of the AB background. **B** Representative images of uninjected and *acvr1b** explants at 12 hpf (equivalent of 4-somite stage) after treatment with triptolide at the indicated stages or with DMSO at 50% epiboly. Scale bar = 200 μm. **C** Length/width ratios of explants shown in (**B**). Each dot represents a single explant from three independent trials, black bars are median values; p < 0.0001, two-sided Mann-Whitney test. **D** Overview of comparisons between published bulk RNA-sequencing experiments from seven developmental stages in three explant conditions (uninjected, *acvr1b**, and *ndr2*) (top) and intact embryos (bottom). Candidate trigger genes were strongly upregulated within the previously determined transcriptional window (5.3–6 hpf). **E** (Top) 180 genes exhibited 'trigger' expression patterns in all explant conditions (light blue); of these, 129 were also upregulated in intact embryos (blue), and 28 showed a sharp increase (Δ > 0.5x) with significant expression levels in intact embryos ( > 5 TPM, dark blue). (Bottom) Heatmap of candidate trigger genes in uninjected, *acvr1b**, and *ndr2* explants. *sumf2* (highlighted in yellow) was selected for further study. **F** Fold-change expression of *sumf1* and *sumf2* transcripts over 4 hpf in explants (solid lines) and embryos (dashed lines) over time. **G** *sumf2/sumf1* expression ratio in explants (solid line) and embryos (dashed line) over time. Note the peak in the *sumf2/sumf1* ratio at 6 hpf, coinciding with the previously determined transcriptional window. Source data are provided as a Source Data file.

Among our candidate genes was *sulfatase modifying factor 2* (*sumf2*). *sumf2* encodes pFGE, a paralog and antagonist of Formylglycine Generating Enzyme (FGE)[65,75], encoded by *sulfatase modifying factor 1* (*sumf1*). As its name implies, FGE converts cysteine residues to formylglycine, a rare but essential post-translational modification required within the active site of all sulfatase enzymes for their activity[75]. Sulfatases catalyze the removal of sulfate groups from a variety of substrates, including lipids, steroids, and glycosaminoglycans, modifying their biological activity[76,77]. Both pFGE (encoded by *sumf2*) and FGE (encoded by *sumf1*) are ER-resident proteins that share high amino acid sequence similarity, with the main distinction that pFGE lacks enzymatic activity[55,75]. Evidence suggests that pFGE binds FGE alone or in complex with its sulfatase substrates[59,65], but it is not clear precisely how pFGE (*sumf2*) antagonizes FGE (*sumf1*) function. Zebrafish FGE (*sumf1*) shares a high degree of amino acid sequence similarity with its human and mouse homologs, including conservation of two essential cysteine residues in the FGE active site (Supplementary Fig. 2A). pFGE (*sumf2*) homologs also showed strong sequence conservation and all lacked the FGE active site (Supplementary Fig. 2B). Zebrafish FGE and pFGE proteins share 46% amino acid identity and 62% similarity (Supplementary Fig. 2C), comparable to their human homologs (48% identity and 62% similarity)[75].

Although other candidate genes exhibited more dramatic expression increases at gastrulation onset, we selected *sumf2* for further study because its function during vertebrate development has never been explored, and its intriguing complementary temporal expression patterns with *sumf1* during gastrulation. In zebrafish embryos and explants, *sumf1* is maternally expressed but exhibits a sharp reduction in transcript levels just prior to gastrulation onset, coinciding with increased *sumf2*[74] (Fig. 1F, Supplementary Fig. 2D). Because *sumf2* and *sumf1* encode a negative and positive regulator of sulfatase activity, respectively, we hypothesized that this peak in the *sumf2/sumf1* ratio at gastrulation onset (6hpf) (Fig. 1G) reduces sulfatase activity and thus increases sulfation of their substrates to trigger C&E movements.

### Excess or loss of sulfatase modifying factors causes C&E defects in zebrafish gastrulae

To investigate the role of sulfatase modifying factors in C&E morphogenesis, we employed both gain- and loss-of-function approaches and performed morphometric analysis at the end of gastrulation (tailbud stage, 10 hpf). We first overexpressed *sumf1* or *sumf2* by mRNA injection into single-cell WT embryos. *sumf1* overexpressing (OE) gastrulae exhibited a significant reduction in anteroposterior (AP) axis length and wider notochords, characteristic of C&E defects, and *sumf2* OE embryos showed similar but milder phenotypes (Fig. 2). Next, we examined gastrulae of the recently characterized *sumf1*[la015919Tg] mutant line (hereafter *sumf1-/-*) harboring a viral insertion in exon 1 that abolishes FGE activity[78]. Homozygous mutant embryos survived to adulthood[78], enabling examination of MZ*sumf1-/-* gastrulae, which also exhibited reduced AP axis length and wider notochords indicative of impaired C&E (Fig. 2).

Because no existing *sumf2* mutant line was available, we used CRISPR to generate a 7 bp insertion in exon 2 of *sumf2* (named *bcm126*), resulting in a premature stop codon (Supplementary Fig. 3A). These mutants also survived to adulthood, enabling us to maintain the line as homozygotes and analyze MZ*sumf2*[bcm126/bcm126] embryos (hereafter MZ*sumf2-/-*). MZ*sumf2-/-* embryos showed a substantial reduction in *sumf2* transcript levels by RT-qPCR (Supplementary Fig. 3C), confirming this as a strong loss-of-function allele. Expression of the closely related *sumf1* was unchanged at early and mid-gastrulation in MZ*sumf2-/-* compared to WT embryos (Supplementary Fig. 3D), suggesting that transcriptional adaptation[79] is not induced by this mutation. ~10% of MZ*sumf2-/-* larvae displayed a tail-curled-up phenotype at 48 hpf and a variable number of MZ*sumf2-/-* adults were undersized and scoliotic (Supplementary Fig. 3E, F), but the remaining larvae and adults had no obvious phenotypes. Although MZ*sumf2-/-* mutant gastrulae did not exhibit C&E defects, *sumf1* OE in this background produced more severe C&E phenotypes than in WT embryos (Fig. 2), supporting an antagonistic role for pFGE (*sumf2*) on FGE (*sumf1*).

To assess whether other gastrulation movements were affected by loss of *sumf1* or *sumf2*, we imaged live MZ*sumf1-/-* and MZ*sumf2-/-* embryos throughout gastrulation and found no significant epiboly defects or delays, with blastopore closure occurring at similar times as in WT embryos (Supplementary Fig. 4A). We also analyzed mesendoderm internalization by quantifying the spread of *sox17*-expressing endoderm cells, which is coupled to mesodermal migration towards the animal pole[80]. MZ*sumf2-/-* gastrulae exhibited normal endoderm spreading, and although MZ*sumf1-/-* gastrulae exhibited reduced spreading at mid-gastrulation, they were not different from WT by late gastrulation (Supplementary Fig. 4B, C). Together, these results demonstrate that altering levels of *sumf1* or/ and *sumf2* specifically disrupts C&E without affecting other gastrulation movements in vivo.

### *sumf1* and *sum2* levels modify the timing of C&E ex vivo

Next, we examined the effects of altered *sumf1* and *sumf2* levels in zebrafish embryonic explants, in which C&E is isolated from other concurrent morphogenetic processes. Surprisingly, neither OE nor deficiency of *sumf1* or *sumf2* impaired the extension of *acvr1b** explants at the equivalent of 4-somite stage (Supplementary Fig. 5A, B), indicating that they are dispensable for C&E per se. To assess whether *sumf1* or *sumf2* levels instead influenced the timing of C&E morphogenesis, we acquired time-lapse recordings of *acvr1b** explants to precisely determine the onset of morphological extension. Extension onset was defined as the time point when a visible tip first emerged from an initially round explant (Fig. 3A, Supplementary Movie 1). Importantly, all explants were staged relative to blastopore closure in age-matched intact embryos of the same genetic background mounted in the same dish. As previously described[36], WT *acvr1b** explants (WT control) and controls co-injected with *superfold-GFP* mRNA (*sfGFP* OE) initiated extension around 8hpf (Fig. 3A, B). Strikingly, extension began significantly later upon *sumf1* OE and significantly earlier upon *sumf2* OE (Fig. 3B). Co-overexpression of *sumf1* and *sumf2* restored the normal time of extension onset (Fig. 3B), demonstrating

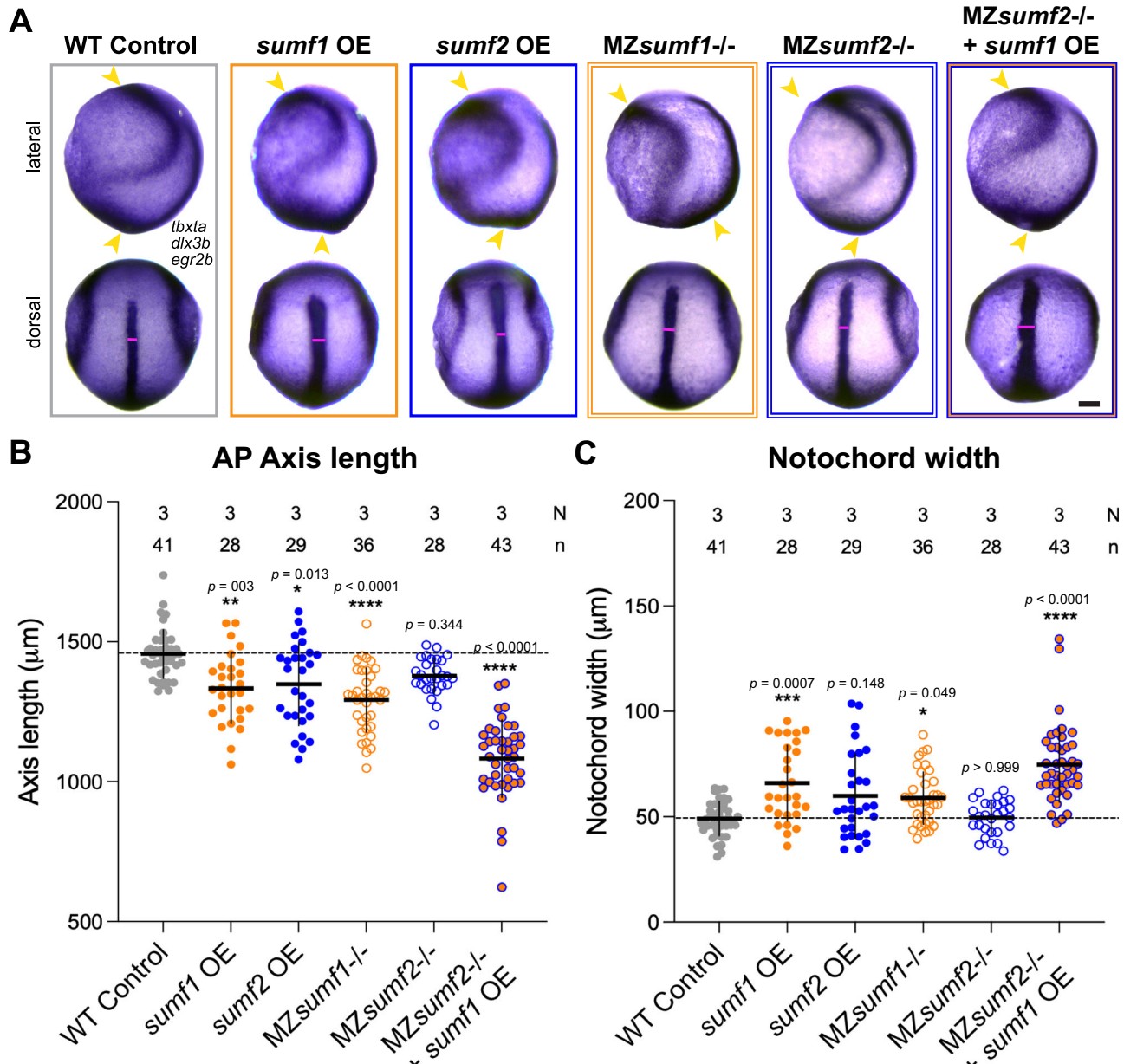

**Fig. 2 | C&E defects in zebrafish gastrulae with altered sulfatase modifying factor levels. A** Representative images of whole mount in situ hybridization (WISH) for *tbxta* (notochord), *dlx3b* (neural plate border) and *egr2b* (rhombomeres 3 & 5) in tailbud stage (10 hpf) zebrafish embryos of the indicated conditions. WT control, MZ*sumf2*-/-, and all OE embryos are on the AB background, MZ*sumf1*-/- embryos are on the TU background. Anterior is up in all images, lateral views are shown on top, dorsal views on bottom. Scale bar = 100 μm. **B, C** Anteroposterior (AP) axis length (**B**, yellow arrowheads) and notochord width (**C**, magenta lines) of embryos depicted in (**A**). Each dot represents a single embryo, N number of independent experiments, n number of embryos. Means and standard deviation are indicated; * $p < 0.05$, * $p < 0.01$, **** $p < 0.0001$ compared to WT control group by Kruskal–Wallis and Dunn's multiple comparisons tests. Source data are provided as a Source Data file.

that the balance between these factors (rather than their absolute levels) determines the time of C&E onset.

In the converse loss-of-function experiments, we found that extension onset was precocious in MZ*sumf1*-/- explants and delayed in MZ*sumf2*-/- explants (Fig. 3A, B), consistent with - but opposite to - *sumf1* and *sumf2* OE phenotypes. MZ*sumf1*-/-; MZ*sumf2*-/- double mutant explants phenocopied the precocious extension of MZ*sumf1*-/- single mutants (Fig. 3B), consistent with pFGE (*sumf2*) modulating C&E timing via its role as an antagonist of FGE (*sumf1*). Notably, explants injected with a morpholino oligonucleotide (MO) against the core PCP component *vangl2*, which does not exhibit a trigger-like temporal expression pattern, exhibited severely impaired C&E morphogenesis

both in vivo and ex vivo (Supplementary Fig. 5A-D)[36], but showed no change in the time of extension onset (Supplementary Fig. 5E). Together, these results demonstrate that the ratio of *sumf1/sumf2* governs the timing of explant extension, distinct from general C&E defects, with higher *sumf1* delaying and higher *sumf2* advancing C&E onset.

To further characterize the dynamics of extension in MZ*sumf1*-/- and MZ*sumf2*-/- explants, we performed automated segmentation-based analysis of time-lapse movies and quantified changes in explant roundness over time (see "Methods"; Fig. 3C). Because explant extension is highly variable and not every explant extends even in control conditions (Supplementary Fig. 5A, B), only explants that reached roundness values below 0.5 (roundness of 1 = perfect circle)

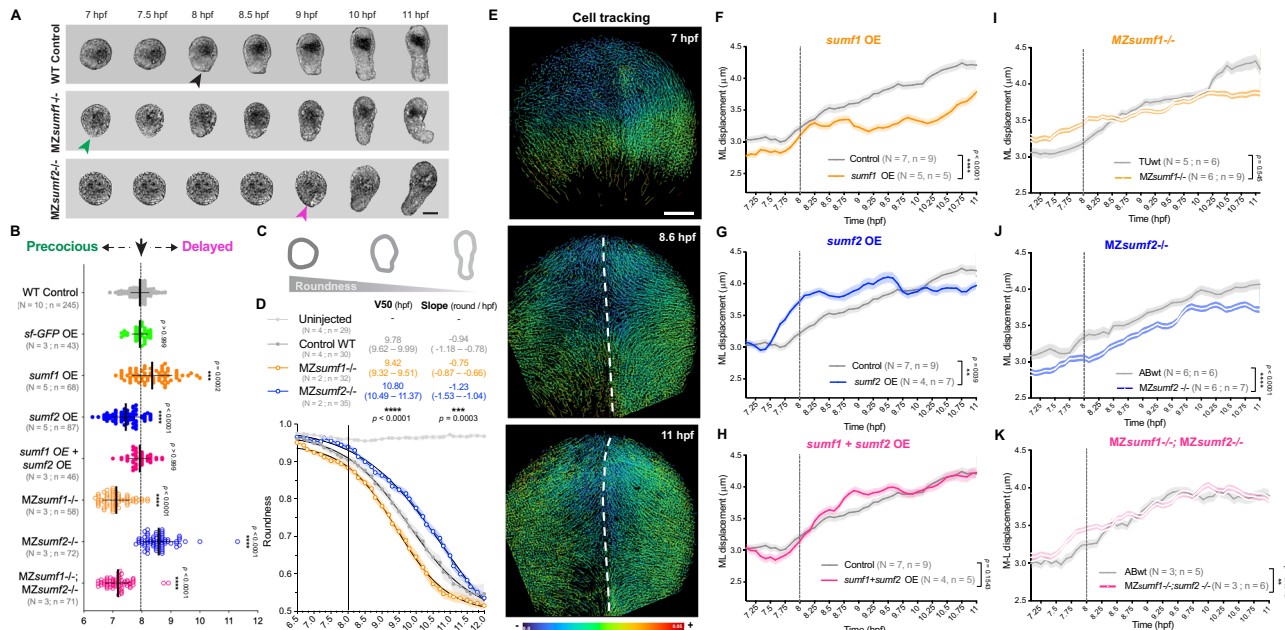

**Fig. 3 | *sumf1* and *sum2* levels control the timing of C&E. A** Representative bright-field images of *acvr1b\** zebrafish explants of the indicated genotypes over time. Black, magenta, and green arrowheads indicate timely, precocious, and delayed onset of extension, respectively. Scale bar = 100 μm. **B** Onset of extension in *acvr1b\** explants of the indicated conditions. Dotted line shows typical extension onset of WT control explants around 8 hpf. Each dot represents a single explant. Means and standard deviation are indicated, N: number of independent experiments, n: number of explants, *** $p < 0.001$, **** $p < 0.0001$ compared with WT control group by Kruskal–Wallis and Dunn's multiple comparisons tests. See Supplementary Movie 1. **C** Time-lapse series of explants were computationally segmented and explant roundness was quantified over time. Diagrams illustrate the progressive reduction in explant roundness during elongation (roundness = 1 indicates a perfect circle). **D** Roundness of explants of the indicated genotypes plotted over time. Means and standard error are indicated, N: number of independent experiments, n: number of explants. Curves were fitted using a sigmoidal model (black lines) and compared using an extra two-sided sum-of-squares F test ($p < 0.0001$). V50 and slope values are shown with 95% confidence intervals, *

$p = 0.003$, **** $p < 0.0001$. All explants were generated from embryos of the AB background except for MZ*sumf1*-/-, which are on the TU background, and MZ*sumf1*-/-;MZ*sumf2*-/-, which are on a mixed AB/TU background.

**E** Representative images of automated nuclear tracking in the dorsal hemisphere of zebrafish gastrulae, starting before C&E onset (7 hpf). Tracks are color-coded by mean speed; dashed lines mark the dorsal midline; scale bar 100 μm. **F–J** ML cell displacement for WT control (gray lines) and *sumf1* OE (**F**), *sumf2* OE (**G**), *sumf1* + *sumf2* OE (**H**), MZ*sumf1*-/- (**I**), MZ*sumf2*-/- (**J**), and MZ*sumf1*-/-; MZ*sumf2*-/- (**K**) embryos. MZ*sumf2*-/- and all OE embryos are on the AB background, MZ*sumf1*-/- embryos are on the TU background, and each condition was compared to WT control embryos of their own genetic background. Each line represents mean and standard error of measured cells for all embryos of a given condition combined, N: number of independent experiments, n: number of embryos. Dotted line shows typical onset of convergence movements in WT control embryos around 8 hpf. ** $p < 0.01$, **** $p < 0.0001$ compared with WT control group by two-sided Wilcoxon signed-rank test. See Supplementary Movie 2. Source data are provided as a Source Data file.

by 12 hpf were included in this analysis. While uninjected explants maintained roundness values close to 1 throughout the gastrulation period, the roundness of *acvr1b\** explants from all analyzed genotypes decreased over time (Fig. 3D). Sigmoidal curves were fitted to the roundness changes of each group, and the V50 (the time at which explants reached 50% of the total roundness change) and slope (reflecting the rate of shape change) were compared using an extra sum-of-squares F test. Significant differences in V50 were observed between groups ($p < 0.0001$), along with significant differences in the slope ($p = 0.003$), indicating changes to both the timing and pace of explant extension upon loss of *sumf2* or *sumf1*. Consistent with previous results, MZ*sumf1*-/- explants began extension precociously, as evidenced by a left-shifted curve and lower V50 compared with WT Controls, but their pace of extension was slower, as indicated by a shallower slope. The opposite was true of MZ*sumf2*-/- explants, whose extension was delayed (higher V50) but faster (steeper slope). This likely explains why explants with different C&E onsets ultimately reach similar degrees of elongation (Supplementary Fig. 5A, B).

### *sumf1* and *sum2* levels modify the timing of C&E in vivo

To determine whether *sumf1* and *sumf2* similarly modify the timing of C&E cell movements in vivo, we acquired confocal time-lapse movies of H2B-scarlet-labeled gastrulae from 6.5–11 hpf and performed automated nuclear tracking (see Methods; Fig. 3E, and Supplementary

Movie 2). To ensure accurate stage matching across experiments, all embryos were staged relative to formation of the second somite, which was readily visible at the end of each movie. Convergence movements were quantified as the mediolateral (ML) displacement of dorsal and lateral cells over time (time frame = 5 minutes) (Fig. 3F–K, and Supplementary Fig. 6). Because convergence movements are reduced within the embryonic midline where extension is more prominent[81], cells within 100 μm of the dorsal midline were excluded from our analysis. In WT control embryos, we observed an upward inflection of ML displacement beginning around 7.75–8 hpf, marking the onset of convergence movements (consistent with previous findings[18,19]), which continued to increase gradually thereafter (Fig. 3F–K, and Supplementary Fig. 6). Upon *sumf1* OE, baseline convergence speed was reduced and, although convergence movements began at the same time as controls, they remained substantially slower (Fig. 3F). *sumf2* OE embryos, on the other hand, had similar baseline convergence speeds as controls but with an earlier inflection point (Fig. 3G). For much of gastrulation, these movements were also faster than those of controls, indicating a likely change in both onset and pace of convergence movements. As in explants, co-OE of both *sumf1* and *sumf2* restored convergence movements to control timing and speeds (Fig. 3H).

Also as in explants, in vivo cell tracking of MZ*sumf1*-/- and MZ*sumf2*-/- mutants revealed opposite effects to their overexpression.

Convergence onset in MZ*sumf1*-/- embryos trended earlier than controls and baseline speed trended higher, although these movements plateaued later in gastrulation (Fig. 3I). Conversely, baseline convergence in MZ*sumf2*-/- mutants was slower than controls with a later inflection point, indicating delayed convergence (Fig. 3J). MZ*sumf1*-/-; MZ*sumf2*-/- embryos displayed an earlier onset of convergence movements (Fig. 3K) as MZ*sumf1*-/- single mutants did, as expected if pFGE (*sumf2*) functions upstream of FGE (*sumf1*). Additional analysis revealed small increases in the persistence of cell movements within MZ*sumf1*-/- gastrulae, but not in other *sumf1* and/or *sumf2* gain- or loss-of-function conditions (Supplementary Fig. 7A), while overall cellular speed remained unchanged in all groups (Supplementary Fig. 7B). Together with our ex vivo results, these data support a model in which pFGE (*sumf2*) counters the activity of FGE (*sumf1*) to control the onset and pace of C&E cell movements during gastrulation.

## Sulfatase modifiers govern C&E via the extracellular sulfatase Sulf1

Because FGE (*sumf1*) and pFGE (*sumf2*) together regulate sulfatase activity levels, we hypothesized that high *sumf2/sumf1* ratios trigger C&E onset by reducing the activity of one or more key sulfatases. Indeed, increased sulfatase activity was shown to disrupt gastrulation morphogenesis in *Xenopus* and sea urchin[64,82–85]. The zebrafish genome encodes 17 sulfatases, 11 of which are expressed during peri-gastrulation stages[74] (Supplementary Fig. 8A). To identify which of these mediates the effect of *sumf1/sumf2* on C&E timing, we overexpressed each of these 11 sulfatases by mRNA injection and performed morphometric analysis at tailbud stage and scored axis phenotypes at 24 hpf. Notably, three of them—the heparan sulfate endosulfatases *sulf1* and *sulf2a*, and the chondroitin sulfatase *arsb*—produced severe AP axis extension defects and increased notochord width, consistent with C&E defects (Supplementary Fig. 8B–E). Unlike *sumf1* and *sumf2* OE, overexpression of these three sulfatases also prevented full extension of *acvr1b** explants at the equivalent of 4-somite stage (Supplementary Fig. 8F, G). However, only *sulf1* OE significantly delayed *acvr1b** explant extension, while the other two had no effect on C&E timing (Fig. 4A, Supplementary Movie 3). Thus, of the 17 zebrafish sulfatases, only Sulf1 was found to affect both C&E morphogenesis and its timing.

To further investigate the role of Sulf1 in zebrafish gastrulation morphogenesis, we examined C&E in MZ*sulf1*sjr9/sjr9 full-locus deletion mutants (Gursimran Kaur Bajwa, Gülsüm Kayman Kürekçi, and Rubén Marín-Juez, manuscript under consideration) (hereafter MZ*sulf1*-/-). MZ*sulf1*-/- embryos exhibited reduced AP axis length at tailbud stage (Fig. 4B-D) and MZ*sulf1*-/- *acvr1b** explants showed impaired extension (Supplementary Fig. 8F, G), indicating that both gain and loss of Sulf1 reduces C&E. Strikingly, MZ*sulf1*-/- *acvr1b** explants exhibited precocious extension (Fig. 4A, Supplementary Movie 3), similar to MZ*sumf1*-/- and *sumf2* OE explants (but opposite to *sulf1* OE). We next quantified the effect of Sulf1 levels on C&E onset in vivo (Supplementary Movie 4). Convergence speed was drastically reduced upon *sulf1* OE (Fig. 4E, Supplementary Fig. 6) while MZ*sulf1*-/- gastrulae exhibited earlier and enhanced convergence movements compared to WT controls (Fig. 4F, and Supplementary Fig. 6). No changes in persistence or cellular speed were detected upon loss or overexpression of *sulf1* (Supplementary Fig. 7). These results highlight Sulf1 as a strong candidate for the key sulfatase controlling C&E onset.

To examine the relationship between Sulf1 and FGE (*sumf1*), we generated *acvr1b** explants co-injected with low doses of each mRNA alone or in combination. The combination of otherwise sub-phenotypic doses of *sumf1* and *sulf1* synergistically delayed and reduced explant extension (Fig. 4G, and Supplementary Fig. 8F, G), consistent with FGE (*sumf1*) enhancing Sulf1 activity. Finally, we tested whether *sumf1* OE could delay explant extension in the absence of *sulf1*. We found that although *sumf1* OE significantly delayed extension

onset in WT explants, it had no effect on the onset of extension in MZ*sulf1*-/- explants, which remained precocious (Fig. 4G). Together, these results implicate Sulf1 as the primary sulfatase through which FGE (*sumf1*) and pFGE (*sumf2*) govern the timing of C&E morphogenesis.

## Heparan sulfate sulfation patterns change with C&E onset and are regulated by *sumf1/sumf2*

Sulf1 is an extracellular sulfatase that removes 6-*O* sulfate groups from heparan sulfate (HS) polysaccharides, with preference towards highly sulfated domains containing IdoA2S-GlcNS6S (D2S6) and, to a lesser degree, GlcA-GlcNS6S (D0S6) disaccharides[67]. We would therefore expect the inversion of FGE (*sumf1*) and pFGE (*sumf2*) levels at the beginning of gastrulation to decrease Sulf1 activity, leading to increased sulfated HS levels at late gastrulation. Indeed, Alcian Blue staining (at pH ~1)[86] for overall glycosaminoglycan (GAG) sulfation increased dramatically from early (50% epiboly) to late (90% epiboly) gastrulation stages, indicating increased embryo-wide GAG sulfation. Treatment with sodium chlorate, which prevents the formation of the sulfate donor 3'-phosphoadenosine 5'-phosphosulfate (PAPS)[87,88], drastically diminished Alcian Blue staining (Fig. 5A), confirming its specificity.

Because Alcian Blue non-specifically stains all negatively charged GAGs, we next isolated HS from WT zebrafish at early and late gastrulation stages by anion-exchange chromatography and quantified levels of HS disaccharides by hydrophilic interaction liquid chromatography coupled with time-of-flight mass spectrometry (HILIC-Q-TOF-MS), using published methods[89]. From this analysis, we found that total HS levels increased significantly between 50% and 80% epiboly stages (Fig. 5B). Although we observed a slight decrease in the percentage of D0S6 disaccharides at late compared to early gastrulation stages (Fig. 5C), there was no significant change in the total abundance of these 6-*O* sulfated disaccharides (reflecting the observed increase in total HS levels) (Fig. 5D). In contrast, while the relative proportion of D2S6 disaccharides remained unchanged during gastrulation (Fig. 5C), their absolute abundance significantly increased (Fig. 5D). These results indicate that abundance of the preferred Sulf1 substrate D2S6 increases during gastrulation, consistent with our hypothesis that the peak *sumf2/sumf1* ratio at gastrulation onset leads to reduced Sulf1 activity.

We next investigated how loss of sulfatase modifying factors affects HS disaccharide composition at late gastrulation (80% epiboly, 8.5hpf). As expected for loss of the key sulfatase activator FGE (*sumf1*), the proportion of unsulfated disaccharides (D0A0) was reduced while proportions of the sulfated disaccharides D0S6 and D2S6 were increased in MZ*sumf1*-/- compared with WT (TUwt) gastrulae (Fig. 5E). We also observed changes in HS disaccharide composition upon loss of pFGE (*sumf2*). Strikingly, the proportion of D2S6 disaccharides was reduced in MZ*sumf2*-/- compared with WT (ABwt) gastrulae (Fig. 5F), consistent with increased sulfatase activity in these mutants. The proportion of D0S6 disaccharides, on the other hand, was increased in MZ*sumf2* mutants, which may reflect compensatory activity of sulfotransferases or other HS modifiers. Importantly, loss of FGE (*sumf1*) and pFGE (*sumf2*) produced opposite effects on the predominant substrate of Sulf1 during gastrulation, supporting their roles as positive and negative regulators, respectively, of Sulf1 activity. The observation that D2S6 disaccharides also increased throughout gastrulation in WT embryos suggests an important role for HS sulfation in C&E morphogenesis.

## HSPGs mediate the effects of sulfatase modifiers on C&E timing

Sulfatases modify a large number of biological substrates, including lipids, steroids, and multiple GAGs[90]. To test whether sulfated heparan sulfate proteoglycans (HSPGs) influence C&E morphogenesis and its timing, we experimentally decreased and increased sulfated HS levels

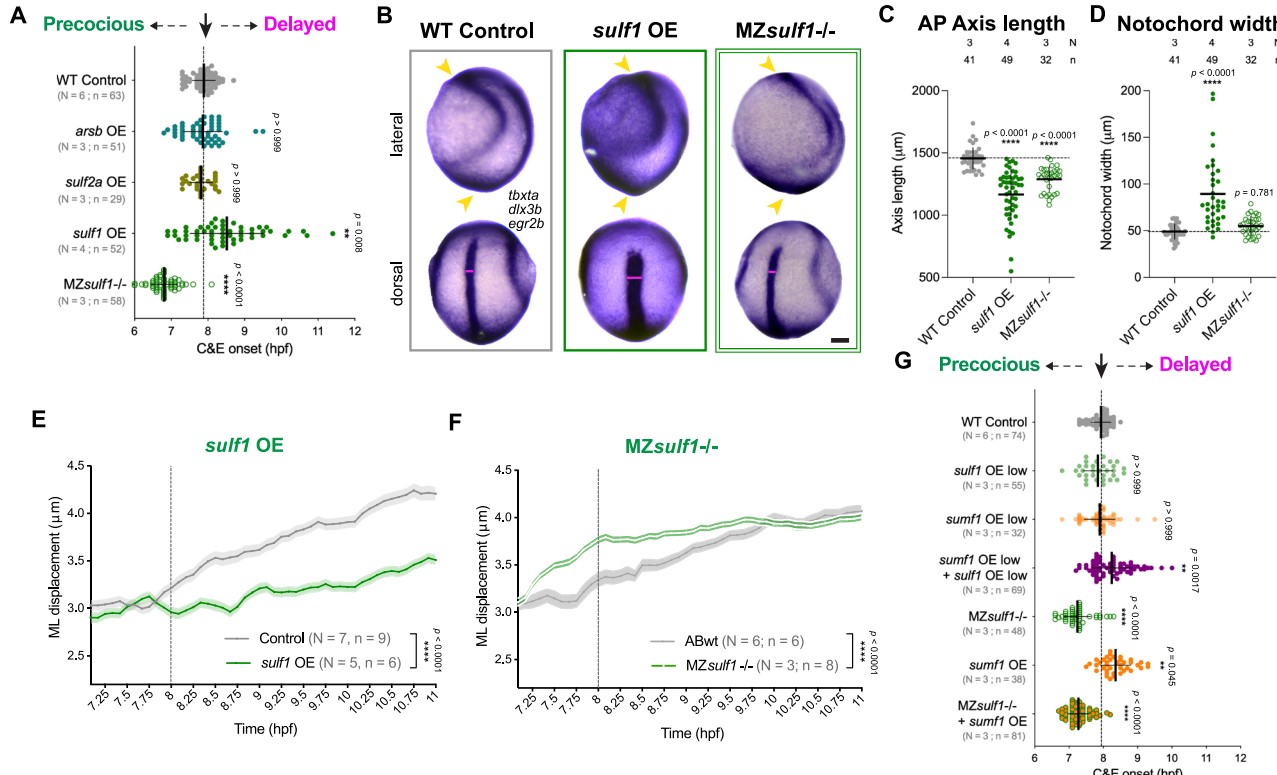

**Fig. 4 | The extracellular sulfatase Sulf1 governs C&E and its timing. A** Onset of extension in *acvr1b** zebrafish explants of the indicated conditions. Dotted line shows typical extension onset of WT control explants around 8 hpf. Each dot represents a single explant. Means and standard deviation are indicated, N: number of independent experiments, n: number of explants. *** *p* < 0.001, **** *p* < 0.0001 compared with WT control group by Kruskal–Wallis and Dunn's multiple comparisons tests. See Supplementary Movie 3. **B** Representative images of WISH for *tbxta* (notochord), *dlx3b* (neural plate border) and *egr2b* (rhombomeres 3 & 5) in tailbud stage (10 hpf) zebrafish embryos of the indicated conditions. Anterior is up in all images, lateral views are on top, dorsal views on bottom. Scale bar 100 μm. (**C, D**) Anteroposterior (AP) axis length (**C**) and notochord width (**D**) were quantified as in Fig. 2 from embryos depicted in (**B**). Each dot represents a single embryo. Means and standard deviation are indicated. **** *p* < 0.0001 compared with WT control group by Kruskal–Wallis and Dunn's multiple comparisons tests. **E, F** ML cell displacement from automated nuclear tracking (as in Fig. 3) in WT control, *sulf1* OE (E) and MZ*sulf1*-/- (**F**) zebrafish gastrulae. Means and standard error are indicated. N number of independent experiments, n number of embryos. **** *p* < 0.0001 compared with WT control group by two-sided Wilcoxon signed-rank test. **G** Onset of extension in *acvr1b** explants of the indicated conditions. Dotted line shows typical extension onset of WT control explants around 8 hpf. Each dot represents a single explant. All explants and embryos are of the AB background. Means and standard deviation are indicated, N number of independent experiments, n number of explants. ** *p* < 0.01, **** *p* < 0.0001 compared with WT control group by Kruskal–Wallis and Dunn's multiple comparisons tests. See Supplementary Movie 4. Source data are provided as a Source Data file.

within embryos and explants. To inhibit overall sulfation, including of GAGs like HS, we treated dome stage embryos (4.3 hpf) with two different doses of sodium chlorate (200 mM and 50 mM), which we previously showed reduced Alcian Blue staining in zebrafish gastrulae (Fig. 5A). We observed severe AP axis shortening and cyclopia in 24 hpf embryos treated with the higher dose, and milder axis defects at the lower dose (Fig. 6A). A dose-dependent effect on C&E was also observed at tailbud stage (Fig. 6B-D), demonstrating that sulfation is required for proper C&E morphogenesis (as previously reported[91–93]). Although explants treated with the higher dose were not viable, treatment with the lower dose of sodium chlorate dramatically reduced extension of *acvr1b** explants (Supplementary Fig. 9A, B). Sodium chlorate treatment also significantly delayed extension onset in both WT control and, importantly, in MZ*sumf1*-/- explants in which extension otherwise occurred early (Fig. 6E, Supplementary Fig. 9C). This finding strongly suggests that loss of *sumf1* induces precocious C&E by enhancing sulfation.

In a complementary experiment, we increased levels of sulfated HS by injecting the extracellular space of 256-cell stage embryos with 10 ng/ml HS (or water as a control) (Fig. 6F). As previously reported[94], HS-injected embryos exhibited dorsalized phenotypes (Supplementary Fig. 9D), likely reflecting increased FGF signaling[95]. Notably, while increased HS did not affect the extension of *acvr1b** explants at the

equivalent of 4-somite stage (Supplementary Figs. 9A, B), their extension onset was significantly earlier than control explants (Fig. 6G, and Supplementary Fig. 9E). HS injection also advanced extension onset in MZ*sumf2*-/- explants (Fig. 6G, and Supplementary Fig. 9E), indicating that increased HS levels are sufficient to trigger precocious C&E ex vivo and can override the delay caused by *sumf2* deficiency. Taken together, these results support a model in which *sumf2* expression at gastrulation triggers the timely onset of C&E cell movements by reducing Sulf1 activity and consequently increasing sulfation of HSPGs.

## Discussion

Morphogenesis requires precise temporal coordination of cell behaviors to ensure proper tissue shape and function, however, the molecular mechanisms governing its timing remain poorly understood. In this study, we uncover a novel molecular mechanism controlling the timing of C&E morphogenesis. After establishing that transcription is required at gastrulation onset for C&E, we identified the sulfatase modifying factors FGE (*sumf1*) and pFGE (*sumf2*) as key temporal regulators. Our data support a model (Fig. 7) in which, prior to gastrulation, maternally deposited FGE (*sumf1*) activates sulfatases enzymes, including the extracellular sulfatase Sulf1. At gastrulation onset, *sumf1* and *sumf2* transcript abundance inverts, and the antagonistic effect of

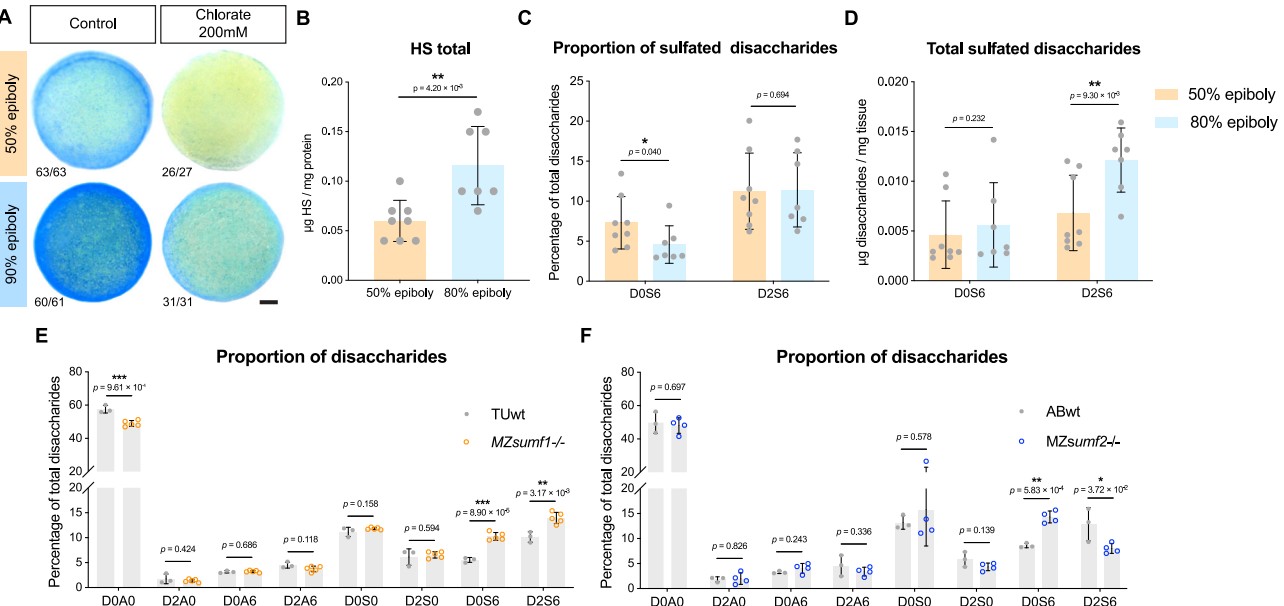

**Fig. 5 | Heparan sulfate sulfation patterns change during gastrulation and are regulated by *sumf1*/*sumf2*. A** Representative images of AB WT zebrafish early (50% epiboly, 5.3 hpf) and late (90% epiboly, 9 hpf) gastrulae stained with Alcian Blue for sulfated GAGs. Alcian Blue staining was diminished in zebrafish embryos treated at dome stage (4.3 hpf) with 200 mM sodium chlorate. Fractions indicate the number of embryos with the depicted staining pattern over number of embryos examined. Scale bar 100 µm. **B**–**D** HILIC-Q-TOF-MS analysis of early and late AB WT zebrafish gastrulae showing the total amount of heparan sulfate (HS) (**B**), the percentage of D0S6 and D2S6 HS disaccharides (**C**), and the total amount of D0S6 and D2S6 disaccharides (**D**) at the stages indicated. **E, F** HILIC-Q-TOF-MS analysis of late zebrafish gastrulae (80% epiboly, 8.5hpf) showing the percentage of each HS disaccharide in MZ*sumf1*-/- versus TU WT (**E**) and in MZ*sumf2*-/- versus AB WT (**F**) genetic backgrounds. Plots show mean and standard deviation, with each dot representing an independent clutch of 100 embryos, **** $p < 0.0001$, *** $p < 0.001$, ** $p < 0.01$, * $p < 0.05$, two-sided Mann-Whitney test. Source data are provided as a Source Data file.

pFGE (*sumf2*) on FGE (*sumf1*) reduces Sulf1 activity. Consequently, increased levels of sulfated HSPGs promote and/or permit C&E morphogenesis.

## Developmental roles of sulfatase modifying factors

This is (to our knowledge) the first report of a role for sulfatase modifying factors in vertebrate gastrulation. In humans, mutations in *SUMF1* cause multiple sulfatase deficiency (MSD), a rare and fatal autosomal recessive disorder characterized by lysosomal dysfunction, developmental delay, neurodegeneration, and skeletal defects including scoliosis, facial dysmorphism, and growth retardation[55,96]. Mouse *Sumf1*-/- models exhibit early postnatal lethality[97], and MZ*sumf1*-/- zebrafish recapitulate some MSD features such as cranial malformations and early growth retardation[78]. However, these fish can survive to adulthood, suggesting the presence of alternative mechanisms of sulfatase activation in this species[78]. Indeed, *Escherichia coli*, *Caenorhabditis elegans*, and *Sacchromyces cerevisiae* possess sulfatase genes but lack a *sumf1* homolog, suggesting the existence of an alternative formylglycine generating enzyme in these organisms[56]. Whether a similar alternative enables MZ*sumf1*-/- zebrafish to survive is unknown.

The role of *sumf2*, on the other hand, had not been studied in vertebrate development. Like MZ*sumf1*-/- mutants, we found that MZ*sumf2*-/- fish can reach adulthood and some individuals exhibit scoliosis (Supplementary Fig. 3E, F). This indicates that although pFGE (*sumf2*) has a significant role in the timing of C&E morphogenesis, it is not ultimately essential for life in zebrafish. Whether its loss is compatible with life in other vertebrate species remains to be tested. Although *Sumf1* orthologs are present in prokaryotes, *Sumf2* is restricted to eukaryotes and has been lost in several metazoan clades, including arthropods[55,75,96,98]. pFGE (*sumf2*) and FGE (*sumf1*) amino acid sequences are highly similar, but pFGE (*sumf2*) lacks formylglycine generating activity due to the absence of an active site[55,59,60] (Supplementary Fig. 2). It was shown in human tissue

culture that pFGE (*sumf2*) reduces FGE (*sumf1*) enzymatic activity[60,65], which is thought to occur through direct physical binding and possible interactions with sulfatase enzymes[59,65]. This antagonism is consistent with our findings in zebrafish that MZ*sumf2*-/- phenotypes are exacerbated by *sumf1* OE (Fig. 2) and co-overexpression of both factors offsets the effects of their individual overexpression (Fig. 3). Further, MZ*sumf1*-/-; MZ*sumf2*-/- double mutant explants phenocopy the precocious C&E observed in MZ*sumf1*-/-, indicating that *sumf1* is epistatic to *sumf2* (Fig. 3), consistent with a model in which pFGE (*sumf2*) governs gastrulation cell movements in its capacity as a FGE (*sumf1*) inhibitor.

## Sulf1 and HSPGs regulate gastrulation cell movements

Our data implicate Sulf1 as the primary sulfatase affecting C&E timing downstream of FGE (*sumf1*) and pFGE (*sumf2*) regulation. Sulf1 catalyzes the removal of 6-*O*-sulfation preferentially from D2S6 disaccharides of heparan sulfate (HS)[67], the highly sulfated disaccharide unit that increases in abundance between early and late zebrafish gastrulation (Fig. 5). Notably, this function is also served by the extracellular sulfatases Sulf2a and Sulf2b. However, *sulf2b* OE did not cause pronounced gastrulation phenotypes, and although *sulf2a* OE induced C&E defects in both embryos and explants (Supplementary Fig. 8), it did not (unlike *sulf1* OE) alter the onset of explant extension (Fig. 4). Although these sulfatases target the same substrate, evidence indicates that human Sulf1 and Sulf2 act on distinct polysaccharide substrates[99], which may also explain the differential effects of *sulf1* and *sulf2a/b* in zebrafish gastrulae.

While the contribution of other sulfatases cannot be excluded, our finding that *sumf1* overexpression no longer affects C&E onset in the absence of *sulf1* strongly implicates Sulf1 as the key sulfatase regulator of C&E timing. Notably, the proportion of D2S6 disaccharides was significantly increased and decreased in MZ*sumf1*-/- and MZ*sumf2*-/- embryos, respectively (Fig. 5), changes that are consistent with their known effects of sulfatase activity. The proportion of D0S6

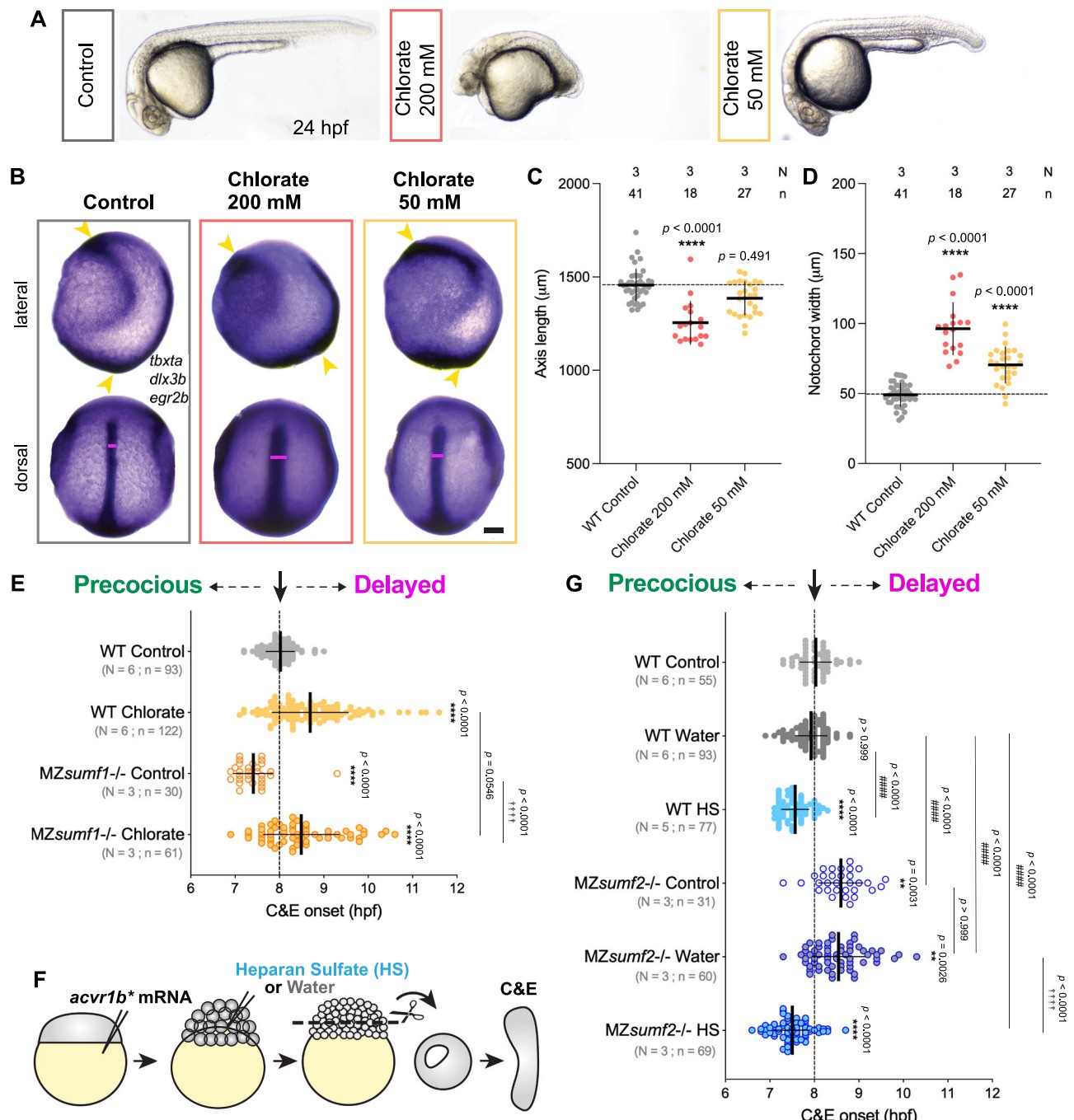

**Fig. 6 | HSPGs mediate the effect of sulfatase modifying factors on C&E timing.** **A** Representative images of 24 hpf zebrafish embryos treated at dome stage (4.33 hpf) with 200 mM or 50 mM of sodium chlorate. **B** Representative images of WISH for *tbxta* (notochord), *dlx3b* (neural border) and *egr2b* (rhombomeres 3 & 5) in tailbud stage (10 hpf) control and sodium chlorate treated embryos shown from lateral (top) and dorsal (bottom) views. Scale bar 100 μm. **C**, **D** AP axis length (**C**) and notochord width (**D**) (as in Fig. 2) of embryos depicted in (**B**). Each dot represents a single embryo. Means and standard deviation are indicated. **** *p* < 0.0001 compared with WT control embryos by Kruskal–Wallis and Dunn's multiple comparisons tests. **E** Onset of extension in *acvr1b\** explants of the indicated conditions. Dotted line shows typical extension onset of WT control

explants around 8 hpf. Each dot represents a single explant. Means and standard deviation are indicated. N number of independent experiments, n number of embryos. ****/ †††† *p* < 0·0001 compared with WT and MZ*sumf1*-/- control groups, respectively, by Kruskal–Wallis and Dunn's multiple comparisons tests. **F** Diagram of extracellular heparan sulfate (HS) or water (control) injections in 256-cell embryos prior to explantation (modified from[36]). **G** Explant extension onset as in (**E**). ** *p* > 0.01; ****/ ####/ †††† *p* < 0.0001 compared with WT, WT + water, and MZ*sumf2*-/- + water groups, respectively, by Kruskal–Wallis and Dunn's multiple comparisons tests. All explants and embryos are of the AB background. Source data are provided as a Source Data file.

disaccharides was similarly increased in MZ*sumf1* and, counter-intuitively, in MZ*sumf2* mutants (Fig. 5). This likely reflects the complex interplay of multiple sulfatases and their various modifiers, as the abundance and sulfation patterns of HS chains are ultimately determined by the activity of many distinct glycosyltransferases,

heparinases, sulfatases and sulfotransferases, among others[100]. Interestingly, *xylosyltransferase 1* (*xylt1*), whose activity initiates HS biosynthesis[101], was also among our candidate trigger genes upregulated at gastrulation onset (Fig. 1), making it an especially compelling candidate for further investigation.

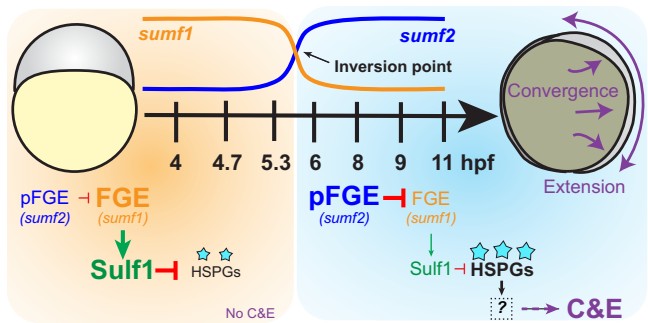

**Fig. 7 | A model for how sulfatase modifying factors regulate the timing of C&E morphogenesis.** Before gastrulation (orange), maternally expressed FGE (*sumf1*) activates Sulf1, maintaining low levels of 6-*O*-sulfation (cyan stars) on HSPGs. At the onset of gastrulation (light blue), an inversion in *sumf1* and *sumf2* transcript abundance (inversion point) increases the antagonistic action of pFGE (*sumf2*) on FGE (*sumf1*), thereby reducing Sulf1 activity. This shift leads to increased 6-*O*-sulfation of HSPGs, generating a permissive environment for C&E morphogenesis via an unknown downstream mechanism.

HSPGs and the sulfatases that modify them have long been known to regulate gastrulation morphogenesis. For example, elimination of HS chains or loss of HSPG core proteins glypicans and syndecans caused gastrulation defects and/or shortened axes in *Xenopus*, zebrafish, and sea urchin gastrulae[86,91,92,102–106]. Although our study does not address the HSPG core protein(s) responsible for the observed effects on C&E timing, glypicans and syndecans with established roles in C&E (like *gpc4/kny*[104]) are good candidates. Notably, our manipulations affecting HSPG sulfation tended to produce milder phenotypes that loss of core proteins like Gpc4, consistent with sulfation as a modifier of their activity. Indeed, manipulations that decrease sulfation levels – including overexpression of *sulf1* or *sumf1* mRNA and injection of purified sulfatases into the blastocoel - also disrupted gastrulation in *Xenopus* and sea urchin[64,82–84]. Interestingly, loss of sulfatases also caused gastrulation defects[84,107,108], suggesting that a correct balance of sulfatase activity is required for proper C&E. Our data provide a possible explanation for this: altered timing of C&E movements. We found that *sulf1* overexpression both disrupts and delays C&E, but also that both precocious and delayed C&E onset ultimately cause C&E defects (i.e., shorter and wider embryonic axes) in intact embryos. This raises the possibility that previously reported C&E defects upon gain or (especially) loss of sulfatase function could be the result of altered timing (discussed further below).

*sulf1* deficiency, however, not only alters the timing of C&E but also prevents full explant extension (Fig. 4, Supplementary Fig. 8). This discrepancy in phenotypic severity between Sulf1 and its key regulator (FGE (*sumf1*)) may reflect partially independent regulation of sulfatases during zebrafish gastrulation. As discussed above, C&E defects resulting from altered sulfatase activity are relatively mild in zebrafish, and the mutant embryos examined here can ultimately survive to become fertile adults. This suggests the existence of compensatory mechanisms that support continued primary body axis elongation after gastrulation is complete.

### Temporal dynamics of gastrulation cell movements

Although it is intuitive that delayed C&E ultimately manifests as a C&E defect in vivo, it is less clear how precocious convergence movements (as seen in MZ*sulf1*-/-, MZ*sumf1*-/-, and *sumf2* OE embryos) lead to similar phenotypes (Fig. 3 and Fig. 4). We speculate that all gastrulation cell movements – epiboly, internalization, and C&E – must be coordinated in both space and time to properly shape the nascent body axes. If C&E movements are accelerated, they become out of sync with other concurrent cell movements, disrupting axis extension. Indeed, although MZ*sumf1*-/- embryos displayed a transient delay in

mesendoderm internalization (Supplementary Fig. 4) we did not observe major changes in the timing of the other gastrulation movements in MZ*sumf1*-/- and MZ*sumf2*-/- embryos. This could explain why morphological C&E defects are detected in MZ*sumf1*-/- mutant embryos but not explants (Fig. 2 and Supplementary Fig. 5), in which C&E occur in the absence of epiboly and internalization. A similar phenomenon was reported in *Drosophila* gastrulae, in which both slowed and accelerated germ band extension desynchronized morphogenesis between the three germ layers[109,110]. Although we cannot rule out a role for sulfatase modifying factors (particularly FGE (*sumf1*)) in the regulation of other morphogenetic processes, our data indicate that *sumf1* and *sumf2* regulate C&E timing without gross changes in other gastrulation cell movements.

It is not yet clear to which degree sulfatase activity alters the onset of cell movements, their pace, or both. For example, either accelerated cell movements or their precocious onset could conceivably manifest as early onset of morphological extension within our explants. Indeed, *Drosophila* embryos with loss- and gain-of function mutations in Serotonin signaling components reportedly exhibit slowed and accelerated cell movements driving germ band extension, respectively, without a change in their time of onset[109–111]. However, our explant extension and in vivo cell-tracking analysis revealed apparent changes in both the onset and speed of convergence movements in zebrafish gastrulae with altered *sumf1*, *sumf2*, and *sulf1* levels, suggesting that multiple aspects of timing are affected (Fig. 3 and Fig. 4). Interestingly, we found that MZ*sumf1*-/- explants extend precociously but also more slowly, suggesting that early C&E onset is not simply a consequence of accelerated C&E (Fig. 3D). This is consistent with our analysis of convergence movements in MZ*sumf1*-/- embryos, in which cell movements began faster but then plateaued at slower speeds than controls (Fig. 3I). The precise changes in cell behaviors upon altered sulfatase activity, and their contributions to observed C&E phenotypes, remain areas of interest for future study.

### How do HSPGs govern morphogenetic timing?

We observed an increase in the levels of sulfated HS (Sulf1 substrates) during late gastrulation when C&E are ongoing (Fig. 5) and demonstrated that both decreased and increased levels of sulfated HS were sufficient to alter C&E timing (Fig. 6). Importantly, the phenotypes associated with *sumf1*, *sumf2*, and *sulf1* perturbations reflect disrupted morphogenesis in the absence of obviously altered cell fate specification, as embryos still give rise to derivatives of all three germ layers, consistent with observations in *Xenopus* and sea urchin gastrulation[82,86]. Although we cannot exclude the possibility that the alterations in C&E timing are secondary to subtle patterning defects or changes in the timing of cell fate choices, our data do not support this as the primary mechanism by which HSPGs regulate C&E timing. Instead, we speculate that modulation of cell signaling and/or extracellular matrix (ECM) are more likely candidates.

Cell surface and extracellular HSPGs play key roles in multiple morphogen signaling pathways. For example, HSPGs modulate ligand diffusion and availability[67,83,112,113] and receptor binding[67,83,114] of morphogens with known roles in C&E, including FGF, Wnt, BMP, and Nodal. HSPGs also regulate PCP signaling through binding of noncanonical Wnt ligands (like Wnt11)[102,104,114] and membrane localization of Dishevelled[83,102,103], which is required for PCP activity[115,116]. The structural features of HS chains, particularly their sulfation patterns, are critical for these functions as they determine ligand-binding affinity and selectivity[100,117,118]. For instance, highly sulfated HS domains, which contain the D2S6 disaccharide, were found to bind the FGF1 ligand with high specificity[119].

The precise role of HSPG sulfation in signaling is context-dependent and changes throughout development. For example, reduced sulfation inhibits and enhances signaling by Wnt8 and Wnt11, respectively[83,84,120,121], and HSPGs from different developmental stages

have different capacities to bind FGF ligands[122,123]. We hypothesize that an increase in sulfated HS during gastrulation may kickstart C&E by modulating the degree and dynamics of HSPG-mediated signaling, for example, by enhancing C&E-promoting FGF signaling and/or dampening C&E-inhibiting BMP signaling. HSPGs can also regulate physical features of the ECM including its composition and stiffness[124–127], which can directly influence migratory cell behaviors ([124,128,129] Whether such roles in morphogen signaling and/or ECM underlie the effect of sulfatase modifiers on C&E timing will be an exciting topic for future studies.

## Methods

All zebrafish experiments were performed in compliance with the Baylor College of Medicine Institutional Animal Care and Use Committee (IACUC), Animal Welfare Assurance number D16-00475. All use of recombinant nucleic acids was performed according to a protocol approved by the Baylor College of Medicine Institutional Biosafety Committee.

### Zebrafish

Adult zebrafish were maintained following established protocols[130] in compliance with the Baylor College of Medicine IACUC, Animal Welfare Assurance number D16-00475. Embryos were obtained through natural mating, and staging was based on established morphology[131]. Fish were chosen from their home tank to be crossed at random, and embryos were randomly chosen from the clutch for injection and inclusion in experiments. WT and mutant embryos were collected in parallel in a period not longer than 10–15 min to minimize developmental differences and raised in egg water at 28.5 °C under identical conditions. Experiments on WT embryos were conducted in either the AB or TU background, depending on the background of the mutant line used, which were obtained from the Zebrafish International Resource Center (ZIRC) via the Baylor College of Medicine zebrafish facility. The mutant lines used were *sumf1*[laO15919Tg] (obtained from ZIRC, ZFIN ID: ZDBALT-120806-11568), *sulf1*[sjr9] (obtained from the University of Montreal)(Gursimran Kaur Bajwa, Gülsüm Kayman Kürekçi, and Rubén Marín-Juez, manuscript under consideration), *sumf2*[bcm126] (this study, described further below), and *oep*[tz257104] (obtained from Washington University School of Medicine). *oep-/-* embryos were rescued to viability by injection of 50 pg of *oep* mRNA[132] and raised to adulthood, then intercrossed to generate MZ*oep-/-* embryos for explantation. MZ*sumf1-/-; MZsumf2-/-* double mutants were generated by crossing single homozygous mutants to obtain double heterozygotes, which were subsequently incrossed.

### *sumf2*[bcm126] generation and genotyping

*sumf2* zebrafish mutants were generated on the AB background using the CRISPR–Cas9 system. A single guide (sg)RNA targeting exon 2 of *sumf2* (5′-GGATGGAGAATCGCCAACAC-3′) was designed using CRISPRscan[133]. sgRNAs were transcribed using T7 RNA polymerase (NEB, M0251S) from DNA templates generated with the forward primer (Fw): gaaattaatacgactcactataGGATGGAGAATCGCCAACACgttttagagctagaaatagc, and the reverse primer (Rv): aaaagcaccgactcggtgccacttttttcaagttgataacggactagccttatttttaacttgctatttctagctctaaaac. 1 µl of gRNA was pre-incubated for 10 min at 37 °C with Cas9 protein (NEB, M0646M) and 300 mM KCl (as described by ref. 134). AB WT Embryos were injected at the single-cell stage with 1 nl of the gRNA–Cas9 complex and cutting efficiency was assessed using a T7 endonuclease I assay[135]. A PCR fragment encompassing the target sequence was amplified from genomic DNA (gDNA) of individual embryos (using the following primers: Fw: AGATGGTGTTTATTCCTGGTGG, Rv: TCCTCTGATACAAAATCCTGGAA) and incubated with T7 endonuclease I (NEB #M0302L) in NEB 10× Buffer 2. Injected F0 embryos were raised and outcrossed to AB WT fish. F1 progeny were genotyped by Sanger sequencing to identify transmitted mutations. A 7 bp insertion in exon 2 of *sumf2* (*sumf2*[bcm126]), resulting in an early stop codon, was

selected (Supplementary Fig. 2A). This mutation introduced an XcmI restriction site, enabling genotyping by digestion of the PCR-amplified target fragment with XcmI (NEB, R0533S) (Supplementary Fig. 3B). Heterozygous F1 carriers were incrossed to produce the F2 generation. Homozygotes F2 fish were raised and incrossed to obtained MZ*sumf2-/-* embryos for experiments.

### RT-qPCR

RNA was isolated from 50 pooled WT and MZ*sumf2-/-* embryos at 30% epiboly (4.66 hpf), 75% epiboly (8 hpf), and 24 hpf from three independent clutches. Total RNA was extracted using TRIzol reagent (Thermo Fisher Scientific, 15596018), purified by sodium acetate precipitation, eluted in 10 mM Tris pH7.5 and treated with Turbo DNAase I (Invitrogen, AM2238) at 37 °C for 30 min. For each sample, 1µg of the RNA was reverse-transcribed using iScript Reverse Transcription Supermix for RT-qPCR (Biorad, #1708840). *sumf2* specific primers were designed to span exon-exon junctions [Exon 1-2 (pair 1) FW: CACAGTGTCTTGTGCAGCAG, RV: AGTTGGAGTTGGTGACAGGA; Exon 3-5 (pair 2) FW: GGCTGAAACATTTGGCTGGA, RV: GGCATCATTCCAGCTGACCT]. *sumf1* specific primers were designed to span exon-exon junctions [Exon 4-5 (pair 1) FW: GGACCAGATTCAACCATACACA, RV: GCTTCTGTAGGAAGTCTGCGTT; Exon 7-9 (pair 2) FW: CTGTATGACATGGTGGGAAATG, RV: GCTGCACACCTGTATCTGTAGC]. *rpl13a-1* was used as a housekeeping gene for normalization [FW: TCTGGAGGACTGTAAGAGGTATGC, RV: AGACGCACAATCTTGAGAGCAG]. qPCR was performed in technical triplicates using SsoAdvanced Universal SYBR Green Supermix (Biorad, #1725270). Relative expression levels were calculated using the $2^{-\Delta\Delta Ct}$ method.

### Amino acid sequence alignment

Amino acid sequence alignment and visualization of human, mouse and zebrafish FGE (SUMF1) and pFGE (SUMF2) orthologues were performed using Jalview2.11.5.1[136]. Multiple sequence alignment was generated with Clustal Omega, and sequence identity and similarity were calculated using the BLOSUM62 substitution matrix.

### Preparation and microinjection of mRNA

All mRNAs were transcribed using the SP6 mMessage mMachine Kit (Fisher Scientific, AM1340) and purified using Bio-Rad Microbiospin columns (Bio-Rad, 7326250). Single-celled embryos were placed in agarose molds (Adaptive Science Tools, I-34) and injected with 0.5-1 nl volumes using pulled glass needles (Fisher Scientific, 50-821-984). Doses of mRNA per embryo were as follows: 0.5 pg *acvr1b*[*69], 10 pg *ndr2*[137], 50-100 pg s*umf1*, 100 pg *sumf2*, 500 pg *sfGFP* (a gift from Dr. Jimann Shin), 500 pg *sulf2b*, 500 pg *sulf2a*, 500 pg *gnsb*, 50-500 pg *sulf1*, 500 pg *sgsh*, 500 pg *arsb*, 500 pg *gnsa*, 500 pg *ids*, 500 pg *galns*, 500 pg *arsa*, 500 pg *sts*, 100 pg *mem-GFP*[11], and 100 pg *H2B-scarlet* (a gift from Dr. Lance Davidson). Templates for *sumf1*, *sumf2*, and all sulfatases were generated by Gibson cloning[138] each of their open reading frames (synthesized by Twist Biosciences) into a PJS2 vector linearized with EcoRI. Requests for these reagents can be addressed to the corresponding author. Injection of 2 ng MO4-*tri/vangl2*[139] was carried out as for synthetic RNA.

### Whole mount in situ hybridization (WISH)

*tbxta* (brachyury, t), *dlx3b*, *egr2b* (krox20), and *sox17* antisense riboprobes were transcribed using NEB T7 RNA polymerase (NEB, M0251s) and labeled with digoxigenin (DIG) NTPs (Sigma/Millipore, 11277073910). WISH was performed according to[140]. Embryos were fixed overnight in at 4 °C 4% PFA in PBS, rinsed in PBS + 10% tween-20 (PBT), and dehydrated into methanol. Following rehydration into PBT, embryos were pre-incubated for at least two hours in hybridization buffer with 50% formamide (in 0.75 M sodium chloride, 75 mM sodium citrate, 0.1% tween 20, 50 mg/mL heparin (Sigma), and 200 mg/mL tRNA) at 70 °C, and hybridized overnight at 70 °C with antisense probes

(1-5 ng/mL) in hybridization buffer. Samples were washed gradually into 2X SSC buffer (0.3 M sodium chloride, 30 mM sodium citrate), and then gradually from SSC to PBT. Samples were blocked at room temperature for several hours in PBT with 2% goat serum and 2 mg/mL bovine serum albumin (BSA), then incubated overnight at 4 °C with anti-DIG antibody (Roche #11093274910) at 1:5000 in block. After extensive washes in PBT, embryos were rinsed in staining buffer (PBT + 100 mM Tris pH 9.5, 50 mM MgCl$_2$, and 100 mM NaCl) and developed in BM Purple AP substrate (Roche) until the desired staining intensity was achieved. Staining was stopped with 10 mM EDTA in PBT before imaging.

### Blastoderm explants

Blastoderm explants were prepared as described by ref. 141. Briefly, uninjected, *acvr1b*\*-, or *ndr2*-injected embryos were dechorionated at the 256-cell stage using pronase (Roche; 1 ml of 20 mg/ml stock in 15 ml 3× Danieau's solution). At the 512-cell stage, ~1/3 of the most animal blastoderm cells were excised using Dumont #55 forceps (Fisher Scientific, NC9791564) on an agarose-coated plate containing 3× Danieau's solution. Explants were allowed to heal briefly before being transferred into agarose-coated 6-well plates containing explant medium [Dulbecco's modified eagle medium with nutrient mixture F-12 (Gibco 11330032) containing 2.5 mM L-glutamine, 15 mM HEPES, 3% newborn calf serum (Invitrogen 26010−066), 50 units/mL penicillin, and 50 mg/mL streptomycin (10,000 U/mL pen-strep at 1:200, Gibco 15140163)]. Explants were incubated at 28.5 °C until sibling embryos from the same genetic background reached the desired developmental stage.

### Alcian Blue staining

Embryos were fixed in PFA 4% overnight, extensively washed in PBS, and incubated in Alcian Blue staining solution pH 1 (0.2% Alcian Blue 8 GX (Sigma # A5268), 50% ethanol, ~0.1 N HCl) for 48 h at room temperature, protected from light. Samples were then gradually rehydrated into PBS, post-fixed in 4% PFA, washed with PBS, and rinsed in 2% KOH. Embryos were cleared through a graded series of glycerol in 2% KOH (20%, 40%, 60%) and stored in 80% glycerol in 2% KOH for imaging and long-term storage.

### Transcription inhibitor treatment

1 µM Triptolide (Sigma # T3652) or an equivalent volume of DMSO was added to the media of blastoderm explants in agarose-coated 6-well plates at 50% (4.7 hpf), shield (6 hpf), 70% (7.5 hpf) or 80% (8.5 hpf). 0.5 µM Flavopiridol (Selleck, S1230) or an equivalent volume of DMSO was added to the media of blastoderm explants at 50% (4.7 hpf) and washed-out twice with 0.3x Danieau solution, before incubation with fresh explant medium.

### Sodium chlorate treatment

Dome stage (4.33 hpf) embryos or blastoderm explants were treated with 200 or 50 mM of Sodium Chlorate (VWR # 7775-09-9) in egg water or explant medium.

### Heparan Sulfate injections

Uninjected or *acvr1b*\*-injected embryos were dechorionated at the 64-cell stage using pronase (Roche; 1 mL of 20 mg/mL stock in 15 mL 3× Danieau's solution). Embryos were transferred to agarose-coated plates containing cubical depressions (Adaptive Science Tools, PT-1) filled with 0.3× Danieau's solution. At the 256-cell stage, embryos were injected with 2 nL of 5 ng/mL heparan sulfate (Sigma-Aldrich, H7640) or nuclease-free water into the extracellular space.

### Microscopy

Live embryos injected with *H2B-mScarlet* and *mem-GFP* mRNA were manually dechorionated and mounted in 0.3-0.35% low-melt agarose (Thermo Fisher Scientific, 16520100) in glass-bottomed 35 mm Petri dishes (Fisher Scientific, FB0875711YZ) for imaging using a Nikon ECLIPSE Ti2 inverted confocal microscope equipped with a Yokogawa W1 spinning disk unit, PFS4 camera, and 405/488/561 nm lasers (emission filters: 455/50, 525/36, 605/52). Temperature was maintained at 28.5 °C during imaging using a Tokai Hit STX stage top incubator. Images were acquired using Nikon NIS Elements software. For live time-lapse series, 100 µm z-stacks with a 2 µm step were collected every 5 minutes for 5 hours using a Plan Apo Lambda 20x dry objective lens. Live blastoderm explants were mounted in rounded chambers made in 1% low-melt agarose in glass-bottomed dishes containing explants medium and imaged in the same scope. For live time-lapses, 14 µm z-stacks were obtained with a 2 µm step every 10 minutes for 6 hours using a Plan Apo Lambda 10x dry objective lens. Images of WISH and alcian blue-stained embryos and live embryos, larvae and explants were taken with a Nikon Fi3 color camera on a Nikon SMZ745T stereoscope. To monitor gastrulation progression in brightfield, live embryos were placed in 1.5% methylcellulose in egg water from the sphere stage and imaged periodically from (4 hpf) to tailbud stage (10 hpf) using a Nikon color camera and stereoscope as above. Embryos were staged according to WT controls.

### Image analysis

ImageJ/Fiji was used to visualize and measure all microscopy data sets.

**Morphometric analyses**. During analysis, researchers were kept unaware of the conditions of all image data using the blind_renamer Perl script (https://github.com/jimsalterjrs/blindanalysis) (blindanalysis: v.1.0.) prior to analysis. To measure the length/width ratios of explants, the length of a segmented line drawn along the midline of each explant (accounting for curvature) was divided by the length of a perpendicular line spanning the maximal width of the explant. Tailbud morphometries were performed in whole-mount embryos staged using *egr2b* expression at future rhombomere 3. Notochord width was quantified in dorsal-view images as the mediolateral extent of the *tbxta* expression domain at the midline. Anteroposterior (AP) axis length was measured in lateral-view images from the anterior to posterior boundaries of the *dlx3b* expression domain. To quantify the extent of endoderm migration toward the animal pole, embryos were imaged laterally at shield, 75% epiboly, and 90% epiboly stages and the spread of *sox17*-positive cells was measured on the ventral side.

**Explant onset of extension**. The onset of explant extension was assessed in a blinded manner as the time point when a visible tip first emerged from an initially rounded explant. If an explant protrusion extended toward or away from the objective in a way that interfered with determining its onset of extension, it was excluded from the analysis. Explants were staged according to sibling intact embryos from the same genetic background cultured in the same plate.

**Explants shape analysis**. Explant time-lapse movies were saved as image sequences (one image per time point) and segmented using Cellpose 2.0[142] implemented in Python. Using FIJI, each individual explant ROI was overlaid onto its corresponding image, and explant roundness was quantified using the standard shape descriptor, where a value of 1 indicates a perfect circle and decreasing values reflect explant elongation. To distinguish changes in the timing or dynamics of extension from defects in C&E morphogenesis, only explants that reached roundness values below 0.5 by 12 hpf were included in the analysis.

**Cell tracking analysis**. Automated nuclear tracking was performed using the ImageJ TrackMate7 plugin[143] in the dorsal hemisphere (encompassing dorsal and lateral cells) of zebrafish gastrulae injected with *H2B-scarlet* mRNA. TrackMate generated color-coded trajectories and measurements of mediolateral (ML) displacement. Embryos were

staged relative to formation of the second somite. To minimize noise from reduced convergence movements near the dorsal midline, cells within 100 μm of the midline were excluded from the analysis. Average displacement in the X (ML) dimension was calculated per time frame, smoothed using a sliding window of four time points, and plotted over time using GraphPad Prism 10. For persistence and speed analyses, only cells that were tracked for at least 15 minutes (three consecutive time frames) were included. Persistence of motion was quantified as the ratio between the linear distance traveled by a cell and the total length of its migration path. Straight-line speed was calculated as the total linear displacement of a cell divided by the total tracking time. Instantaneous velocity was defined as the cell's spatial displacement between two consecutive frames divided by the frame interval (5 min).

## GAG isolation from zebrafish embryos
To compare GAG composition between stages, sibling AB WT embryos were collected and flash frozen in liquid nitrogen at 5.3 (8 biological replicates) and 8.5 hpf (7 biological replicates), with each replicate containing 100 embryos. To compare GAG composition between WT and mutants, MZ*sumf1*-/- (on a Tuebingen background, 5 biological replicates), MZ*sumf2*-/- (on an AB background, 4 biological replicates), and control WT embryos of Tuebingen and AB strains, respectively (3 biological replicates each) were collected and flash frozen at 8.5 hpf, with each replicate containing 100 embryos. Whole embryos were homogenized and lysed in 0.5% CHAPS lysis buffer (50 mM HEPES, 120 mM NaCl, 2 mM EDTA, pH 7.4) containing a protease inhibitor cocktail (Roche). 50 μL of cell lysate was set aside for protein quantification via BCA assay. Homogenates were diluted 1:10 in a wash buffer (50 mM sodium acetate, 200 mM NaCl, 0.1% Triton X-100, pH 6.0) and incubated with Pronase (0.4 mg/ml, Sigma) overnight at 37 °C with mild agitation. The product was centrifuged (4,000 xg, 20 minutes) then passed through a DEAE-Sephacel (Cytiva) column equilibrated in 50 mM sodium acetate buffer, pH 6.0, containing 200 mM NaCl, and desalted using a PD-10 desalting column (Cytiva), and lyophilized to dryness overnight.

## HS depolymerization and HILIC-Q-TOF-MS disaccharide analysis
For HS disaccharide analysis, purified GAGs were digested with 2 mU each of heparin lyases I, II, and III (IBEX) for 16 hours at 37 °C in a buffer containing 40 mM ammonium acetate and 3.3 mM calcium acetate, pH 7. Samples were dried in a centrifugal evaporator then aniline-tagged, as previously described[89]. Briefly, $[^{12}C_6]$aniline (17 μL, Sigma-Aldrich) and 17 μL of 1 M NaCNBH$_3$ (Sigma-Aldrich) freshly prepared in dimethyl sulfoxide:acetic acid (7:3, v/v) were added to each sample. Reactions were carried out at 37 °C for 16 hours and then dried in a centrifugal evaporator. Dried samples were resuspended in LC-MS grade water and spiked with an equimolar mixture of $[^{13}C_6]$aniline-tagged HS disaccharide standards (20 pmol, Iduron) prior to HILIC-Q-TOF-MS analysis. HILIC-UPLC was performed on a Waters Acquity UPLC system (Waters Corporation). Separation was achieved on a Waters Acquity UPLC Amide BEH column (2.1 mm × 150 mm, 1.7 μm) maintained at 40 °C. Mobile phases consisted of (A) acetonitrile and (B) 50 mM ammonium formate in water at pH 4.4 (Ammonium Formate Solution - Glycan Analysis, Waters). For HS disaccharide analyses, the following gradient program was used: 0–5 min, 90% A; 5–48 min, 90–67% A; 48–60 min, 67% A; 60–65 min, 67–90% A; and 65–70 min, 90% A. The flow rate was 0.5 mL/min in all runs. The injection volume was 2 μL. The UHPLC system was coupled to a Waters Synapt XS Q-TOF mass spectrometer equipped with an electrospray ionization source operated in negative ion mode. The following source parameters were used: source temperature 80 °C, desolvation temperature 250 °C, cone gas flow 50 L/h, desolvation gas flow 1000 L/h, capillary voltage 2.0 kV, sampling cone 35 V, and source offset 4.0 V. Data were acquired in resolution mode from 200 to 1000 m/z. MassLynx v4.2 software (Waters Corporation) was used for molecular feature extraction and data processing. Samples were quantified using isotopically labeled $[^{13}C_6]$aniline-tagged internal disaccharide standards and normalized to total protein, as measured by BCA.

## Statistical analysis
Number of embryos or explants (n) and independent experimental replicates (N) for embryo or explant studies are stated in graphs. All experiments were performed at least in triplicate. GraphPad Prism 10 software was used to perform statistical analyses and generate graphs for all data analyzed. Datasets were tested for normality prior to analysis and statistical tests were chosen accordingly. The statistical tests used for each data set are noted in figure legends, and all tests used were two-sided. For explant roundness changes over time, curves were fitted using a Boltzmann sigmoidal model. Curve comparisons were performed using an extra sum-of-squares F test, comparing a model in which parameters were shared across datasets with a model in which parameters were allowed to vary between groups.

## Reporting summary
Further information on research design is available in the Nature Portfolio Reporting Summary linked to this article.

## Data availability
Raw RNA-sequencing data were previously published[71] and are available in NCBI Gene Expression Omnibus under accession number GSE246158. Raw data for LC-MS analysis of glycosaminoglycans generated in this study are available at GlycoPOST[144] under project ID GPST000676. All image and gene expression analysis data generated for this study are available in the Supplementary and Source Data files. Source data are provided with this paper.

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

## Acknowledgements

We thank Dr. Lila Solnica-Krezel for sharing plasmids and WISH probes, the BCM Center for Comparative Medicine for taking excellent care of our fish, and the Zebrafish International Resource Center for preserving and distributing fish lines used here and by countless members of the community. Thanks also to all members of the Williams lab for their help and feedback on this project, and to Drs. Maria Cecilia Cirio and Lance Davidson for their thoughtful comments on the manuscript. This work was supported by NIH/NICHD grants R00HD091386 and R01HD104784 to M.K.W. The glycosaminoglycan disaccharide analyses were supported by NIH grant R35GM150736 to R.J.W., and those performed at the CCRC were partially supported by NIH grant R24GM137782 to Parastoo Azadi. G.K.B. was supported by a Bourses d'excellence (Université de Montréal) and a FRQ Doctoral Scholarship. CIHR grants (PJT-178037, PJT-204048) and FRQS J1 and J2 awards provided support for R.M.J. S.G. and C.C. were partially supported by CPRIT RP210227 and RP200504, NIH/NCI P30 shared resource grant CA125123, NIH/NIEHS P42 ES027725 and P30 ES030285. Data analysis was performed on the HPC cluster that is managed by the Biostatistics and Informatics Shared Resource (BISR) and supported by an NIH S10 Shared Instrument Grant S10-OD032185, NCI P30-CA125123 and Institutional funds from the Dan L Duncan Comprehensive Cancer Center and Baylor College of Medicine.

## Author contributions

A.S.C. and M.K.W. conceived of the project. A.S.C. and M.K.W. performed zebrafish experiments, R.J.W. and A.B. performed HS disaccharide profiling, and R.M.J. and G.K.B. generated the *sulf1* deletion line. S.G. and C.C. performed bioinformatic analysis. A.S.C and M.K.W. wrote the original manuscript. All authors reviewed the manuscript.

## Competing interests

The authors declare no competing interests.
