## [Peer Review file · Nature Communications]

Sulfatase modifying factors control the timing of zebrafish convergence and extension morphogenesis

Corresponding Author: Dr Margot Williams

Version 0:

Reviewer comments:

Reviewer #1

(Remarks to the Author)
NCOMMS-25-85681

Title: Sulfatase modifying factors control the timing of zebrafish convergence and extension morphogenesis

Convergence and extension (C&E) movements are essential for proper embryogenesis and must occur at a precise time during gastrulation. However, the mechanisms controlling this timing is unknown. The primary goal of this study was to elucidate the molecular factors involved in C&E timing using a zebrafish explant model previously developed by the lab. Through bulk RNA-sequencing analysis, the authors identified two genes involved in sulfatase modification- sumf1 and sumf2- with reciprocal expression dynamics at the onset of C&E movements in gastrulation. The authors followed with overexpression and knock-out experiments in explants and in vivo to confirm it is the ratio of Sumf1 and Sumf2 that is required for C&E onset in gastrulation. Further analysis revealed Sumf1 and Sumf2 affect the activity of the sulfatase Sulf1. The authors were able to establish a mechanism wherein low Sumf1/high Sumf2 prevents the activity of Sulf1 leading to more sulfated heparan sulfate proteoglycans presence in the embryo at the onset of C&E movements during gastrulation. This study is well-done; manuscript is well prepared with robust data. I have but a few minor comments to be addressed:

1. More explanation about how the cut-offs were determined for identifying the 28 candidates from a list of 129 (lines 140-142) would be useful.
2. The manuscript did not address whether sumf2^{-/-} embryos have a phenotype, although presumably since only MZsumf2^{-/-} was described, they did not. Is this true? It is curious that only the MZ have a phenotype considering Sumf2 is not maternally deposited. Is there perhaps a paralog of Sumf2 that is compensating or another biological reason? Please clarify.
 - a. Related to this point, the manuscript describes sumf2^{-/-} adults were scoliotic and undersized (line 195-196) however the referenced supplementary figure labeled these fish as MZsumf2^{-/-}. Please clarify or relabel.
3. The manuscript does show the ratios of Sumf1/Sumf2 is important for onset of C&E in gastrulation. This argument may be strengthened by the addition of an additional condition wherein both Sumf1 and Sumf2 are absent via double mutants (sumf1^{-/-}; sumf2^{-/-}), knockdown of sumf1 in MZsumf2^{-/-} or vice versa to see if CE is still affected.
4. On page 20, line 432, the authors refer to "craniofacial abnormalities" present in some MZsumf2^{-/-} zebrafish. These "abnormalities" are not evident from the current images in Supplementary Fig. 2D-2E. Please either remove this phrase or offer additional images depicting the craniofacial phenotype in the zebrafish and/or Alcian blue staining to visualize the craniofacial structures.
5. What is the sumf1 expression level in the MZsumf2^{-/-} mutant embryos? How does it change before vs. during gastrulation? Answering these questions via qPCR would offer additional background for the MZsumf2^{-/-} + sumf1 OE experiments.

Minor critiques:

1. Scale bar is missing in Figure 1B.
2. Scale bar is missing in Figure 3A.

3. To make Figure 3A more accessible to readers that cannot easily discern red from green, the authors are encouraged to use different colors to depict arrows.

Reviewer #2

(Remarks to the Author)

This work presents novel findings relating to the role of sulfatases in regulating the timing of gastrulation onset in zebrafish embryos. While we know a relatively large amount about the molecular mechanisms governing the control of cell intercalation events that act to produce convergence and extension movements during gastrulation, relatively little is known about how these events are coordinated in time during development. This work provides the first evidence of a novel mechanism governing this process that acts at the level of controlling the production of heparan-sulfate proteoglycans. These results are novel and interesting, and will set the stage for future studies investigating the role of post-translational modifiers in controlling the timing of developmental events. The discussion places the observations in a broader context of what is known about HSPGs in the regulation of known events critical to the progression of gastrulation: the regulation of morphogen activity for multiple signalling pathways. I have some suggestions for further improving the manuscript before it is ready for publication:

- 1) The characterisation of 'trigger' genes associated with gastrulation onset, and are expressed within the critical window of gene expression required for C&E onset is interesting. The authors select two genes to follow up with, a provide a supplementary table with the other gene expression changes. It would be informative to dig a little deeper into these additional gene expression changes and summarise them within the main text of the paper.
- 2) Many of the experiments involve the quantification of elongation onset in embryonic explants. This was performed in a blinded manner by visually determining when the first protrusion is generated from an explant. Is this a reliable method for quantification? There could be issues with instances where explants protrusion towards or away from the objective, rather than laterally. It would be helpful to discuss these potential caveats or look for automated shape analysis that could be employed to validated the findings.
- 3) Fig. 3. It is not clear whether the traces shown for ML displacement are representative traces from single movies, or combined statistics from multiple independent movies. If the former, can the authors include the additional data in supplementary materials?
- 4) Resumably a major impact of HSPG modification would be on components of the ECM, that might in turn explain the defects in gastrulation morphogenesis. Could the authors discuss this alongside their speculation over the potential interaction with known signalling pathways?

Reviewer #3

(Remarks to the Author)

In this manuscript, Cervino et al address the fundamental, yet poorly understood, question of how the timing of morphogenetic movements are controlled during gastrulation. The authors identify a role for the ratio of sumf2 and its paralog sumf1 in controlling the onset and dynamics of CE movements in zebrafish. They further identify the Sulf1, an extracellular sulfatase that modifies heparan sulfate proteoglycans (HSPGs), as a key effector by which sumf1 and sumf2 regulate tissue movements.

The manuscript addresses an exciting open question in biology and identifies an unexpected role for sulfatase enzymes in regulating morphogenesis in zebrafish. This is a very interesting avenue of research, however, in my view, the mechanism and interpretation of the phenotypes remain rather unclear, with important assumptions from the authors or knowledge gaps requiring, in my view, further testing/work. Moreover, the *in vivo* data is, in my view, more consistent with an important regulatory role for sulfatase activity in regulating CE movements, rather than a timer mechanism (which I nonetheless find very exciting!). See my detailed points below.

1. How clear is the notion that sumf1 and sumf2 have opposite roles on sulfatase activity in zebrafish? Is this consistent with the domain organization and homology of both enzymes to well-studied FGE and pFGE enzymes in other organisms?
2. Do the levels of sulfated HSPG levels change systematically, and as expected, in the context of overexpression and downregulation of either enzyme and w/the expected kinetics? This will also clarify how much the differential expression pattern of these 2 enzymes would be expected to be sufficient to tip sulfatase activity *in vivo*...
3. A key point in the manuscript is that the shift in the ratio between sumf1/sumf2 triggers CE movements – how is the expression of sumf2 regulated?

These points are at the core of the mechanism and require additional work in my view.

It is unclear to me why the sumf1 and sumf2 manipulations would display CE defects in vivo, as shown by the authors. Based on the timer model, I would have only expected a delay/accelerated onset (more consistent w/the observations in the explants, but not with the in vivo work). Instead, the defects ML movements and axis extension suggest rather a regulatory role in CE movements. In line with this, upon sumf1 OE in vivo, the difference in CE onset is rather modest, with a much clearer delay in mediolateral displacement during CE (as also pointed out by the authors). This is also consistent with the phenotypes of Sul1 and the treatments with sodium chlorate.

4. Additional characterization of the tracks for persistence, directionality is also needed to better understand these cellular defects in CE.

5. The authors should track ML displacements in explants, where no obvious macroscopic defect in CE is detected? Are there only timing differences or also changes in cell movement in this context? Could the in vivo phenotype be a result of a shorter window for CE? (that would still not explain why the embryos that start CE earlier, also shows defects though)

6. How do these phenotypes compare with well-established regulators of CE? Would these not produce both a change in onset and in the speed of ML movements during CE, just like the manipulation of sulfation?

7. How does sulfation act to regulate CE movements? Are there differences in morphogen signaling for instance (e.g. BMP or FGF range, levels, dynamics) or rather in the distribution of PCP components? This is also important to understand whether this is a timer mechanism or a regulator of CE.

8. In vivo, are other morphogenetic movements affected by manipulations of sulfatase activity? This is another important point to better characterize the phenotypes observed in vivo.

9. A final important point in my view is to characterize cell type proportions upon OE/knockout of the sumf1/sumf2/sul1.

Overall, based on the in vivo data, however, it is, in my view, hard to conclude that this is a timer mechanism for CE rather than an additional and important regulator of cell behaviors driving CE. This is the center premise of the manuscript, so additional work is, in my view, required in this regard to clarify the core of the author's model.

Version 1:

Reviewer comments:

Reviewer #1

(Remarks to the Author)

The authors have fully addressed my comments and I have no additional comments. Thank you for the opportunity to review this work.

Reviewer #2

(Remarks to the Author)

The authors have satisfactorily address all of my previous concerns.

Reviewer #3

(Remarks to the Author)

I much appreciated the clarifications and additional experiments provided by Cervino, et al in the revised manuscript, which has improved the clarity of their manuscript and strengthened their findings substantially in my view.

I would still request a few more clarifications and analysis from the authors before publication, namely:

1. I am not convinced that the rate of mediolateral movements would be affected in Sumf1 and Sumf2 mutants and OE in vivo simply due to a change in onset alone – even if the total duration of these movements is affected as indicated by the authors in the rebuttal, again the expectation here would be that they proceed at regular speed. Thus, I think it is important to directly and explicitly discuss these different possibilities in the Discussion section. In addition, please provide the quantification of the epiboly rates and larvae length discussed in the rebuttal for the Sumf1 and Sumf2 in the manuscript.
2. Please move the data from S9 into the corresponding main figure.
3. Please include, for comparison, the expression dynamics for well-known regulators of CE (e.g. vangl2) in the corresponding plots (or as supplementary data).
4. A discussion on the specificity of the timing defects for CE vs other morphogenetic processes would be beneficial, especially considering the slight defects in internalization rate in the MZsumf1^{-/-} embryos.
5. While I agree that gross defects in the cell type proportions are unlikely, an imbalance between axial and paraxial populations in the context of the mutant is still important to address in my view.

Response to Reviewers

NCOMMS-25-85681

Cervino et al.

Sulfatase modifying factors control the timing of zebrafish convergence and extension morphogenesis

Reviewer #1 (Remarks to the Author)

Convergence and extension (C&E) movements are essential for proper embryogenesis and must occur at a precise time during gastrulation. However, the mechanisms controlling this timing is unknown. The primary goal of this study was to elucidate the molecular factors involved in C&E timing using a zebrafish explant model previously developed by the lab. Through bulk RNA-sequencing analysis, the authors identified two genes involved in sulfatase modification- *sumf1* and *sumf2*- with reciprocal expression dynamics at the onset of C&E movements in gastrulation. The authors followed with overexpression and knock-out experiments in explants and in vivo to confirm it is the ratio of *Sumf1* and *Sumf2* that is required for C&E onset in gastrulation. Further analysis revealed *Sumf1* and *Sumf2* affect the activity of the sulfatase *Sulf1*. The authors were able to establish a mechanism wherein low *Sumf1*/high *Sumf2* prevents the activity of *Sulf1* leading to more sulfated heparan sulfate proteoglycans presence in the embryo at the onset of C&E movements during gastrulation. This study is well-done; manuscript is well prepared with robust data. I have but a few minor comments to be addressed:

We thank the Reviewer for their kind words and their appreciation of the significance and rigor of our study.

1. More explanation about how the cut-offs were determined for identifying the 28 candidates from a list of 129 (lines 140-142) would be useful.

Based on the outcome of our time course of Triptolide treatment in explants, we hypothesized that genes required for timely C&E onset would exhibit a “trigger” like pattern, meaning that their expression would increase substantially between 50% epiboly and shield stages. To identify such genes in explants, we sorted for transcripts whose expression did not increase prior to shield stage but then increased significantly and remain elevated at shield stage and beyond. To focus in on genes with potential in vivo relevance, we further selected for genes with similar expression patterns in intact zebrafish embryos but whose 1) increase in expression and 2) expression level were substantial enough to reasonably affect our biology of interest. To this end, we selected an arbitrary cut-off of 1) a 50% increase in transcript level from 50% epiboly to shield stage in intact embryos and 2) an expression level of at least 5 TPM at shield stage. When narrowed down this way, we were left with 28 candidate genes to choose from. This expanded description has been added to the Results section in lines 131-146.

2. The manuscript did not address whether *sumf2*^{-/-} embryos have a phenotype, although presumably since only *MZsumf2*^{-/-} was described, they did not. Is this true? It is curious that only the MZ have a phenotype considering *Sumf2* is not maternally deposited. Is there perhaps a paralog of *Sumf2* that is compensating or another biological reason? Please clarify.

a. Related to this point, the manuscript describes *sumf2*^{-/-} adults were scoliotic and undersized (line 195-196) however the referenced supplementary figure labeled these fish as *MZsumf2*^{-/-}. Please clarify or relabel.

The Reviewer raises a great point. Because *sumf2* is not maternally expressed, generating *MZsumf2*^{-/-} embryos is not strictly necessary. However, because these *Zsumf2*^{-/-} mutants are viable and develop into fertile adults, we chose to maintain this line as homozygotes for practical reasons, namely that it eliminated the need to genotype individual explants, which is infeasible. We did not examine the phenotypes of zygotic mutants during gastrulation or at the tailbud stage, but we did observe larvae with axis defects and adults with scoliosis among *Zsumf2*^{-/-} progeny obtained from F1 incrosses (which were similar to the larval and adult phenotypes observed in *MZsumf2*^{-/-} animals). *sumf1* is the only known paralog of *sumf2*, but because these two genes exhibit antagonistic functions, we don't expect genetic compensation by *sumf1* in *sumf2* mutants. Indeed, as described further in point 5 below, we observe no change in *sumf1* expression levels within *MZsumf2*^{-/-} embryos. Finally, we have corrected the text to reflect that the animals shown in the supplementary figure are in fact *MZsumf2*^{-/-}. We apologize for this oversight.

3. The manuscript does show the ratios of Sumf1/Sumf2 is important for onset of C&E in gastrulation. This argument may be strengthened by the addition of an additional condition wherein both Sumf1 and Sumf2 are absent via double mutants (*sumf1*^{-/-}; *sumf2*^{-/-}), knockdown of *sumf1* in *MZsumf2*^{-/-} or vice versa to see if CE is still affected.

We agree entirely with the Reviewer's comment and have added the analysis of *in vivo* convergence movements in *MZsumf1*^{-/-}; *MZsumf2*^{-/-} embryos. These can be found in the new Figure 3K. We find that convergence movements in double mutants resemble those of *MZsumf1*^{-/-} single mutants, demonstrating that *sumf1* is epistatic to *sumf2*, as would be expected if *sumf2*/pFGE functions by antagonizing *sumf1*/FGE. This description was added to the results section in lines 316-321.

4. On page 20, line 432, the authors refer to "craniofacial abnormalities" present in some *MZsumf2*^{-/-} zebrafish. These "abnormalities" are not evident from the current images in Supplementary Fig. 2D-2E. Please either remove this phrase or offer additional images depicting the craniofacial phenotype in the zebrafish and/or Alcian blue staining to visualize the craniofacial structures.

We agree, and as characterization of skeletal abnormalities resulting from Sumf2 loss of function is beyond the scope of the present manuscript, we have removed the sentence referring to this phenotype.

5. What is the *sumf1* expression level in the *MZsumf2*^{-/-} mutant embryos? How does it change before vs. during gastrulation? Answering these questions via qPCR would offer additional background for the *MZsumf2*^{-/-} + *sumf1* OE experiments.

We thank the Reviewer for this suggestion and have now added a qPCR analysis of *sumf1* expression at pre-gastrulation and late gastrulation stages in WT and *MZsumf2*^{-/-} embryos. These data are now shown in Suppl. Fig. 3D (formerly Suppl. Fig. 2) and demonstrate that *sumf1* expression levels in *MZsumf2*^{-/-} embryos are not different from WT at either time point. A description of these results was added to the results section in lines 201-203.

Minor critiques:

1. Scale bar is missing in Figure 1B.
2. Scale bar is missing in Figure 3A.
3. To make Figure 3A more accessible to readers that cannot easily discern red from green, the authors are encouraged to use different colors to depict arrows.

We appreciate the Reviewer's comments and have made all the suggested corrections.

Reviewer #2 (Remarks to the Author):

This work presents novel findings relating to the role of sulfatases in regulating the timing of gastrulation onset in zebrafish embryos. While we know a relatively large amount about the molecular mechanisms governing the control of cell intercalation events that act to produce convergence and extension movements during gastrulation, relatively little is known about how these events are coordinated in time during development. This work provides the first evidence of a novel mechanism governing this process that acts at the level of controlling the production of heparan-sulfate proteoglycans. These results are novel and interesting, and will set the stage for future studies investigating the role of post-translational modifiers in controlling the timing of developmental events. The discussion places the observations in a broader context of what is known about HPSGs in the regulation of known events critical to the progression of gastrulation: the regulation of morphogen activity for multiple signalling pathways. I have some suggestions for further improving the manuscript before it is ready for publication:

We thank the Reviewer for their positive evaluation of our study and their recommendations to improve it.

- 1) The characterisation of 'trigger' genes associated with gastrulation onset, and are expressed within the

critical window of gene expression required for C&E onset is interesting. The authors select two genes to follow up with, and provide a supplementary table with the other gene expression changes. It would be informative to dig a little deeper into these additional gene expression changes and summarise them within the main text of the paper.

Although we did not provide much detail about these other candidates in the original submission, we agree that these additional genes may be of interest to the readership. First, we provided gene annotation information in Supplementary table 1, including a new tab with GO, KEGG, and INTERPRO terms for all 28 candidate genes. We have further added a discussion of these genes and their functions to the main text in the results section in lines 142-146.

2) Many of the experiments involve the quantification of elongation onset in embryonic explants. This was performed in a blinded manner by visually determining when the first protrusion is generated from an explant. Is this a reliable method for quantification? There could be issues with instances where explants protrude towards or away from the objective, rather than laterally. It would be helpful to discuss these potential caveats or look for automated shape analysis that could be employed to validate the findings.

The reviewer raises an important point. If an explant protrusion extended toward or away from the objective in a way that interfered with determining its onset of extension, it was scored as “out of focus” and excluded from the analysis. We have added a statement in the Methods section to clarify this. In addition, we performed automated shape analysis of WT, *MZsumf1*, and *MZsumf2* mutant explants and plotted their roundness over time, which is now shown in our new Figure 3C. We found that this type of analysis was not suitable for reliably determining the precise onset of extension because the emergence of an extension tip did not substantially alter explant roundness until well after it first became visually apparent. However, this automated did enable comparisons of the dynamics of explant extension across conditions. As expected, these curves were shifted leftward (earlier) and rightward (later) in *MZsumf1*^{-/-} and *MZsumf2*^{-/-} explants, respectively, but the rate of elongation (i.e. the slope of explant roundness over time) also differed between these conditions. This analysis corroborates and strengthens our conclusion that the onset of C&E movements is altered by levels of sulfatase modifiers and revealed additional effects on the pace of C&E. A description of these results was added to the results section in lines 253-268.

3) Fig. 3. It is not clear whether the traces shown for ML displacement are representative traces from single movies, or combined statistics from multiple independent movies. If the former, can the authors include the additional data in supplementary materials?

The plots in Figure 3 are the mediolateral displacement of cells combined from multiple embryos, with the number of analyzed embryos (n) from independent experiments (N) indicated in the graph. Statistical analyses were also performed on these combined traces, comparing each experimental group with its corresponding WT control. Per the Reviewer’s suggestion, we have now also added single-embryo tracks in Figure Suppl. 6.

4) Presumably a major impact of HSPG modification would be on components of the ECM, that might in turn explain the defects in gastrulation morphogenesis. Could the authors discuss this alongside their speculation over the potential interaction with known signalling pathways?

The Reviewer raises an excellent point. We have now added a discussion of potential effects of altered HSPG sulfation on the ECM and how this may contribute to the observed C&E defects in lines 619-622.

Reviewer #3 (Remarks to the Author):

In this manuscript, Cervino et al address the fundamental, yet poorly understood, question of how the timing of morphogenetic movements are controlled during gastrulation. The authors identify a role for the ratio of *sumf2* and its paralog *sumf1* in controlling the onset and dynamics of CE movements in zebrafish. They further identify the *Sulf1*, an extracellular sulfatase that modifies heparan sulfate proteoglycans (HSPGs), as a key effector by which *sumf1* and *sumf2* regulate tissue movements.

The manuscript addresses an exciting open question in biology and identifies an unexpected role for sulfatase enzymes in regulating morphogenesis in zebrafish. This is a very interesting avenue of research, however, in my view, the mechanism and interpretation of the phenotypes remain rather unclear, with important assumptions from the authors or knowledge gaps requiring, in my view, further testing/work. Moreover, the in vivo data is, in my view, more consistent with an important regulatory role for sulfatase activity in regulating CE movements, rather than a timer mechanism (which I nonetheless find very exciting!). See my detailed points below.

We are glad to hear this Reviewer found our study to be interesting and exciting. We are confident that our revisions have improved the manuscript and hope they will satisfactorily address the Reviewer's concerns.

1. How clear is the notion that *sumf1* and *sumf2* have opposite roles on sulfatase activity in zebrafish? Is this consistent with the domain organization and homology of both enzymes to well-studied FGE and pFGE enzymes in other organisms?

We thank the reviewer for raising this point. The opposing roles of SUMF1/FGE and SUMF2/pFGE have been established in human cell lines, and this study represents, to our knowledge, the first functional analysis of *sumf2* in zebrafish. We have added to Figure Suppl. 2 an alignment of the amino acid sequences of zebrafish *sumf1* and *sumf2*, showing high homology of each with their human and mouse homologs. Consistent with the mammalian homologs, zebrafish *sumf2* lacks the two cysteine residues present in the active site of *sumf1*. Discussion of these findings was added to the results section in lines 156-161. We also present strong functional data demonstrating opposing roles on sulfatase function. First, overexpression of *sumf1* delays C&E onset which is 'cancelled out' by co-overexpression of *sumf2*. Second, double *MZsumf1*; *MZsumf2* mutant explants and now embryos (newly added to Figure 3J) resemble *MZsumf1* single mutants, demonstrating that *sumf1* is epistatic to *sumf2*, as we would expect if *sumf2*/pFGE functions by antagonizing *sumf1*/FGE. Finally, we performed HILIC-Q-TOF-MS profiling of HS disaccharides in *MZsumf1* and *MZsumf2* mutant embryos as suggested by the Reviewer in point 2 below. We found that these mutations had opposite effects on the proportion of sulfated D2S6 disaccharides, which are the predominant substrate of Sulf1. Together, the strong conservation with mammalian homologs and our functional experiments support an antagonistic role between *sumf1* and *sumf2* in zebrafish.

2. Do the levels of sulfated HSPG levels change systematically, and as expected, in the context of overexpression and downregulation of either enzyme and w/the expected kinetics? This will also clarify how much the differential expression pattern of these 2 enzymes would be expected to be sufficient to tip sulfatase activity in vivo...

The reviewer raises an important point. To address it, we have profiled HS sulfation by HILIC-Q-TOF-MS in *MZsumf1* and *MZsumf2* mutant embryos at the 80% epiboly time point when C&E is active. The resulting findings have been added to Figure 5E and F, and show that the composition of sulfated disaccharides is indeed altered in both mutants compared with their respective WT controls. We found that *MZsumf1*^{-/-} embryos had an increased proportion of sulfated D2S6 disaccharides, the predominant substrate of Sulf1, while *MZsumf2*^{-/-} embryos had a decreased proportion of this same disaccharide. This demonstrates a significant change in sulfatase activity and downstream HSPG profiles upon altered levels of both sulfatase modifiers. A description of these results was added to the results section in lines 400-412.

3. A key point in the manuscript is that the shift in the ratio between *sumf1*/*sumf2* triggers CE movements – how is the expression of *sumf2* regulated?

We agree entirely that this is an important and interesting question! However, addressing it would likely require an entire additional manuscript's worth of work and is thus, beyond the scope of the current study. We hope to be able to answer this question in the future.

These points are at the core of the mechanism and require additional work in my view.

It is unclear to me why the *sumf1* and *sumf2* manipulations would display CE defects in vivo, as shown by the

authors. Based on the timer model, I would have only expected a delay/accelerated onset (more consistent w/the observations in the explants, but not with the in vivo work). Instead, the defects ML movements and axis extension suggest rather a regulatory role in CE movements. In line with this, upon *sumf1* OE in vivo, the difference in CE onset is rather modest, with a much clearer delay in mediolateral displacement during CE (as also pointed out by the authors). This is also consistent with the phenotypes of *Sulf1* and the treatments with sodium chlorate.

The Reviewer raises an important point. Indeed, our explant assay allows us to detect changes in the onset of C&E (seen in *MZsumf1* and *MZsumf2* mutants) as well as more general C&E defects (seen in *MZsulf1*^{-/-} and chlorate treated explants). Changes in C&E onset can also be detected in intact embryos by nuclear tracking, as the Reviewer noted, even if C&E is not disrupted *per se*. We speculate that defects in axis length and width measured in fixed mutant embryos result not from a general C&E defect but rather from a miscoordination of C&E with other concurrent cell movements. For example, if epiboly finishes on time (which it does in both *MZsumf1* and *MZsumf2* mutants) but C&E is delayed or precocious, these two movements would be out of sync with one another and could therefore manifest as the observed C&E defects. Further speculation on this point can be found in the discussion section in lines 572-574. The fact that these embryos eventually recover to form larvae and adults of normal length further suggests that this is a timing effect, since embryos with general C&E defects (for example, *trilobite* and *knypek* mutants) develop into very short larvae that don't survive beyond ~5 days.

4. Additional characterization of the tracks for persistence, directionality is also needed to better understand these cellular defects in CE.

We thank the reviewer for this suggestion, and have added these cell tracking statistics in the new Suppl. Fig. 7. A description of these results was also added to the results section in lines 316-321.

5. The authors should track ML displacements in explants, where no obvious macroscopic defect in CE is detected? Are there only timing differences or also changes in cell movement in this context? Could the in vivo phenotype be a result of a shorter window for CE? (that would still not explain why the embryos that start CE earlier, also shows defects though)

This is a great idea. We are indeed able to track labeled nuclei within explants (as we showed in PMID 32319426), but convergence movements are very challenging to detect and measure *ex vivo*. In intact embryos, cells converge from the lateral regions of the embryo across the surface of the yolk, resulting in substantial ML displacement that we can easily measure. By contrast, explants have no yolk and thus, although extension movements are readily apparent, convergence movements are not. However, we are able to address this question without nuclear tracking. We performed automated analysis of WT, *MZsumf1*^{-/-}, and *MZsumf2*^{-/-} explant shape and plotted explant roundness over time (shown in our new Figure 3C-D), providing a readout of explant extension dynamics. This curve was shifted earlier and later for *MZsumf1*^{-/-} and *MZsumf1*^{-/-} explants, respectively, corroborating our conclusion that C&E onset changes with loss of sulfatase modifiers. However, these curves also had modest changes in slope, revealing that precocious C&E movements were slightly slower and delayed C&E movements were slightly faster than WT controls. The result is that the period of C&E is the same in all conditions. A description of these results was added to the results section in lines 253-268.

6. How do these phenotypes compare with well-established regulators of CE? Would these not produce both a change in onset and in the speed of ML movements during CE, just like the manipulation of sulfation?

This is an excellent point. To address it, we analyzed the onset of extension in explants cut from *vangl2* MO-injected embryos, which exhibit severe C&E defects due to disruption of planar cell polarity (PCP) signaling. As expected, the overall extension of *vangl2* MO explants was significantly impaired, as we previously described in PMID 32319426. However, we found that their extension onset time was not affected (in the new Suppl. Fig. 5D). This demonstrates that a general C&E defect need not manifest as a change in C&E onset and vice versa, and that regulation of each can be molecularly distinct. A description of these results was added to the results section in lines 248-250.

7. How does sulfation act to regulate CE movements? Are there differences in morphogen signaling for instance (e.g. BMP or FGF range, levels, dynamics) or rather in the distribution of PCP components? This is also important to understand whether this is a timer mechanism or a regulator of CE.

We completely agree that this is an exciting area of research, and we hope to pursue it in the future. However, to answer this question would require months or even years of further experiments that would likely comprise an entire additional manuscript. For this reason, we believe it is beyond the scope of the current study.

8. In vivo, are other morphogenetic movements affected by manipulations of sulfatase activity? This is another important point to better characterize the phenotypes observed in vivo.

Indeed, we feel this study is novel and exciting because manipulation of sulfatase modifier levels affects the timing specifically of C&E without creating a general developmental delay or altering the other gastrulation cell movements of epiboly and ingression. To further support this claim, we have provided a new Suppl. Fig. 4. In panel A, we include representative images of WT, *MZsumf1*^{-/-}, and *MZsumf2*^{-/-} gastrulae showing no major delays in epiboly movements, as all embryos achieve blastopore closure within similar time frames. We also added analyses of endoderm spreading (which, because they are coupled, also indicates mesoderm spreading) in these genetic backgrounds (Suppl. Fig. 4B,C). Although reduced endodermal spreading was observed in *MZsumf1*^{-/-} embryos at mid-gastrulation, the embryos 'caught up' by late gastrulation, reaching levels comparable to WT embryos. Overall, while we cannot rule out minor roles for sulfatase modifying factors in other morphogenetic processes, their loss of function does not severely impair these behaviors during gastrulation. A description of these results was added to the results section in lines 209-217.

9. A final important point in my view is to characterize cell type proportions upon OE/knockout of the *sumf1/sumf2/sul1*.

Based on whole mount in situ hybridization analyses of ectodermal (*dlx3b*, *egr2b*), mesodermal (*tbxta*), and endodermal (*sox17*) marker genes (shown in multiple figures throughout the manuscript), we did not observe obvious changes in germ layer specification or cell type proportions in either loss- or gain-of-function conditions for *sumf1* or *sumf2*. The spatial expression patterns and domain size of these markers were comparable to those in WT embryos, indicating that overall tissue patterning is preserved. Further, these embryos survive to adulthood, demonstrating that all cell types must be present in approximately normal numbers. Although we cannot rule out small changes in cell type proportions, we have no evidence that *sumf1* and *sumf2* play a major role in early cell fate specification.

Overall, based on the in vivo data, however, it is, in my view, hard to conclude that this is a timer mechanism for CE rather than an additional and important regulator of cell behaviors driving CE. This is the center premise of the manuscript, so additional work is, in my view, required in this regard to clarify the core of the author's model.

We are confident that the additional evidence described here strengthens the support for our model that *sumf1* and *sumf2* regulate the timing of C&E during zebrafish gastrulation and satisfactorily address the Reviewers' concerns.

We thank all three Reviewers for their careful assessment of the revised version of this manuscript. Our responses to the remaining concerns raised are below:

Reviewer #3 (Remarks to the Author):

I much appreciated the clarifications and additional experiments provided by Cervino, et al in the revised manuscript, which has improved the clarity of their manuscript and strengthened their findings substantially in my view.

I would still request a few more clarifications and analysis from the authors before publication, namely:

1. I am not convinced that the rate of mediolateral movements would be affected in *Sumf1* and *Sumf2* mutants and OE in vivo simply due to a change in onset alone – even if the total duration of these movements is affected as indicated by the authors in the rebuttal, again the expectation here would be that they proceed at regular speed. Thus, I think it is important to directly and explicitly discuss these different possibilities in the Discussion section.

When completing our first round of revisions, we found that the rate of extension does indeed differ between explants with precocious and delayed C&E onset, such that the total period of C&E does not differ between conditions. Evidence from our nuclear tracking in vivo supports this ex vivo finding, and we accordingly state in the Results and Discussion sections (see lines 452, 475, and 707) that both onset and speed of convergence movements are affected by manipulating levels of *sumf1* or *sumf2*.

In addition, please provide the quantification of the epiboly rates and larvae length discussed in the rebuttal for the *Sumf1* and *Sumf2* in the manuscript.

In Supplemental Fig. 3E, we show that about 90% of *MZsumf2*^{-/-} larvae are indistinguishable from WT, although we have not quantified their length. Supplemental Fig. 4A also shows 3 representative, age-matched embryos for each WT, *MZsumf1*^{-/-}, and *MZsumf2*^{-/-} embryos undergoing epiboly. Although these were not formally quantified, these images show clearly that the rate of epiboly is not affected in these mutants.

2. Please move the data from S9 into the corresponding main figure.

We have moved these data into Figure 5.

3. Please include, for comparison, the expression dynamics for well-known regulators of CE (e.g. *vangl2*) in the corresponding plots (or as supplementary data).

We now show log2fold expression changes for *vangl2* in explants of all 3 conditions and in intact embryos in Supplemental Fig. 5.

4. A discussion on the specificity of the timing defects for CE vs other morphogenetic processes would be beneficial, especially considering the slight defects in internalization rate in the *MZsumf1*^{-/-} embryos.

We indeed found that the reported timing defects are specific to C&E and do not affect the other gastrulation cell movements of epiboly and internalization. This is discussed in lines 693-701.

5. While I agree that gross defects in the cell type proportions are unlikely, an imbalance between axial and paraxial populations in the context of the mutant is still important to address in my view.

I fear that we cannot fully address this concern without performing single-cell RNA-sequencing, which would require substantial time and resources. Because we cannot formally rule out the possibility that altered proportions of different cell types contribute to the observed phenotypes, we state this in the Discussion section (see line 724). However, we also note that our data do not support this as the primary mechanism by which sulfatase modifiers affect C&E timing.